EMBO
Molecular Medicine

# Targeting prominin2 transcription to overcome ferroptosis resistance in cancer

Caitlin W Brown ⓘ, Peter Chhoy ⓘ, Dimpi Mukhopadhyay ⓘ, Emmet R Karner ⓘ & Arthur M Mercurio ⓘ

## Abstract

Understanding how cancer cells resist ferroptosis is a significant problem that impacts ongoing efforts to stimulate ferroptosis as a therapeutic strategy. We reported that prominin2 is induced by ferroptotic stimuli and functions to resist ferroptotic death. Although this finding has significant implications for therapy, specific prominin2 inhibitors are not available. We rationalized that the mechanism by which prominin2 expression is induced by ferroptotic stress could be targeted, expanding the range of options to overcome ferroptosis resistance. Here, we show that that 4-hydroxynonenal (4HNE), a specific lipid metabolite formed from the products of lipid peroxidation stimulates *PROM2* transcription by a mechanism that involves p38 MAP kinase-mediated activation of HSF1 and HSF1-dependent transcription of *PROM2*. HSF1 inhibitors sensitize a wide variety of resistant cancer cells to drugs that induce ferroptosis. Importantly, the combination of a ferroptosis-inducing drug and an HSF1 inhibitor causes the cytostasis of established tumors in mice, although neither treatment alone is effective. These data reveal a novel approach for the therapeutic induction of ferroptosis in cancer.

**Keywords** cancer; ferroptosis; heat shock factor 1; prominin2; therapy
**Subject Categories** Autophagy & Cell Death; Cancer

## Introduction

Elucidating mechanisms that enable cancer cells to resist ferroptosis, a regulated form of non-apoptotic cell death characterized by the accumulation of peroxidated lipids, is critical for understanding the nature of ferroptosis in cancer and exploiting its therapeutic potential (Dixon *et al*, 2012; Yang & Stockwell, 2016; Yang *et al*, 2016). Stimuli that disrupt intracellular glutathione-mediated antioxidant systems can trigger ferroptosis. For example, inhibition of the xCT transporter reduces intracellular cystine, a key precursor of glutathione (GSH). This results in a reduction of available GSH to act as a co-factor for glutathione peroxidase 4 (GPX4), the main protein involved in reducing lipid peroxidation (Dixon *et al*, 2012; Yang *et al*, 2014). The fact that ferroptosis sensitivity varies widely among cancer cell types indicates that some cancer cells can resist the lipid peroxidation damage that is the root cause of ferroptosis (Yang *et al*, 2014). This issue is of profound importance for ongoing efforts aimed at stimulating ferroptosis as a therapeutic approach for cancer (Ou *et al*, 2017; Zhang *et al*, 2019). A paramount example of this problem is the observation that some tumor cells are highly susceptible to GPX4 inhibition while others are not (Yang *et al*, 2014; Hangauer *et al*, 2017).

Recently, we described a novel mechanism of ferroptosis resistance that involves the pentaspanin protein prominin2 (Brown *et al*, 2019). In cells that are resistant to ferroptosis, the expression of prominin2 is induced rapidly by stimuli that increase lipid peroxidation including GPX4 inhibition and detachment from the extracellular matrix. Prominin2 promotes the formation of multivesicular bodies (MVBs) that contain iron-laden ferritin. The export of these MVBs as exosomes reduces the intracellular iron concentration and, consequently, results in the evasion of ferroptosis. A key finding in this study was that depletion of prominin2 expression renders otherwise resistant cells vulnerable to ferroptosis. Although this finding has significant implications for therapy, specific prominin2 inhibitors are not available. We therefore rationalized that the mechanism by which prominin2 expression is induced by ferroptotic stress could be targeted as well, expanding the range of options to reduce ferroptosis resistance.

In this study, we discovered that a metabolite resulting from degraded, peroxidated lipids, 4-hydroxynonenal (4HNE) stimulates *PROM2* transcription by a mechanism that is dependent upon heat shock factor 1 (HSF1). Furthermore, we demonstrate that specific inhibition of HSF1 overcomes resistance to GPX4 inhibition and significantly increases the sensitivity of resistant cancer cells *in vitro* and tumors *in vivo* to ferroptosis. This finding increases our understanding of mechanisms used by cells to resist ferroptosis, and it has potentially valuable implications for improving therapies aimed at stimulating ferroptosis in cancer.

## Results

### Lipid peroxidation induces prominin2 expression

Initially, we sought to determine the nature of the stimulus that induces prominin2 expression in cells that are resistant to

Department of Molecular, Cell and Cancer Biology, University of Massachusetts Medical School, Worcester, MA, USA
*Corresponding author. Tel: +1 508 856 8676; Fax: +1 508 856 1310; E-mail: arthur.mercurio@umassmed.edu

    EMBO Molecular Medicine    13: e13792 | 2021    **1 of 14**

ferroptosis. Treatment of MCF10A cells with the GPX4 inhibitor RSL3 resulted in a time-dependent increase in *PROM2* mRNA expression that was significant at 60 min and peaked at 120 min (Fig 1A). Induction of *PROM2* mRNA was blocked by ferrostatin-1 (fer-1), a specific inhibitor of lipid peroxidation (Fig 1B). Given that ferroptosis is an iron-dependent process, we treated MCF10A cells with exogenous iron in the form of ferric ammonium citrate (FAC), which also resulted in an induction of *PROM2* mRNA. However, this iron-dependent increase in *PROM2* was inhibited by ferrostatin-1 (Fig EV1A) indicating that the ability of iron to induce *PROM2* expression derives from its ability to trigger lipid peroxides and not its independent signaling capacity.

The metabolites that result from the breakdown of peroxidated lipids have significant signaling capacity (Esterbauer *et al*, 1982, 1990). These metabolites include malondialdehyde (MDA), 4-hydroxynonenal (4HNE), 4-hydroxyhexenal (4HHE), 4-oxo-nonenal (4ONE), among others. We observed that exposure of MCF10A cells to 4HNE was sufficient to increase *PROM2* mRNA significantly in the absence of ferroptotic stress (Fig 1C). As expected, ferrostatin-1 pretreatment did not rescue this increase in expression because 4HNE is downstream of the oxidized lipids (Fig 1D). Both RSL3 and 4HNE increased prominin2 protein expression after 1 h of treatment (Fig 1E). In contrast, exposure to neither MDA, 4ONE nor 4HHE significantly impacted the expression of prominin2 (Fig EV1B). Importantly, RSL3 treatment increased the level of 4HNE as evidenced by its ability to increase the formation of 4HNE protein adducts, which can be detected by immunoblotting, similar to the level observed with exogenous 4HNE treatment (Fig 1E). In contrast, exposure to MDA did not significantly impact the expression of prominin2 (Fig 1F). This result is consistent with the fact that 4HNE is a stress-derived aldehyde that can act as a "second messenger" of redox signaling and modulate signaling pathways and transcriptional programs (Codreanu *et al*, 2009; Smathers *et al*, 2011), while MDA oxidizes and cross-links DNA (Niedernhofer *et al*, 2003).

### 4HNE signals through p38 MAP kinase to induce prominin2

4HNE signals by forming Michael adducts with target kinases resulting in their phosphorylation and activation (Amarnath *et al*, 1998). The p38/MAPK cascade has been shown to be a specific target of 4HNE (Zarrouki *et al*, 2007; Zheng *et al*, 2013), and there is evidence that 4HNE forms adducts with p38 that trigger its phosphorylation and activation (Usatyuk & Natarajan, 2004). Consistent with these findings, we observed that 4HNE or RSL3 stimulation of MCF10A cells increased the phosphorylation of p38 MAP kinase at 60 min of treatment, a time frame consistent with its ability to induce prominin2 expression (Fig 2A). We also assessed whether p38 is a direct target of 4HNE by immunoprecipitating phospho-p38 (Tyr180/Thr182) and immunoblotting for the 4HNE adduct. We found that 4HNE was bound to phospho-p38 in 4HNE-treated but not in control cells, indicating that p38 is a direct target of 4HNE (Fig 2B).

To evaluate whether p38 contributes to the induction of *PROM2* expression, cells were treated with either 4HNE or RSL3 in the presence or absence of BIRB, a specific p38 inhibitor. Inhibition of p38 prevented the induction of *PROM2* mRNA expression by either 4HNE or RSL3 (Fig 2C) but BIRB treatment alone did not affect

prominin2 protein levels (Fig EV2C). These results were recapitulated at the protein level in both MCF10A cells and the breast cancer cell line Hs578t, showing an increase in prominin2 that is dependent on the presence of an active p38 (Figs 2D and EV2A). Similar findings were obtained with SB202190, another pan-p38 inhibitor (Fig EV2D). These data demonstrate that 4HNE-stimulated p38 activity drives the increase in prominin2 expression.

A similar experiment was performed to determine the contribution of p38 to ferroptosis resistance. MCF10A and Hs578t cells were treated with BIRB, RSL3, or a combination of the two for 12 h and surviving cells were quantified. Neither BIRB nor RSL3 alone affected cell survival. However, their combined treatment reduced survival by 50%, indicating that p38 signaling is a key component of ferroptosis resistance (Figs 2E and EV2B). Together, these data indicate that cells can resist ferroptosis by activating p38 and, consequently, inducing prominin2.

### The induction of *PROM2* mRNA by 4HNE is mediated by HSF1

The critical issue that arose from our p38 data is how this kinase induces *PROM2* mRNA expression. To address this issue, we utilized a compilation of ENCODE ChIP-seq data to identify transcription factors that bind within the *PROM2* promoter, gene body, or enhancer regions. Searching this database for transcription factors binding to *PROM2* served as an unbiased approach that yielded the heat shock factor 1 (HSF1) as a notably strong *PROM2* interactor (Fig 3A). HSF1 can induce or repress expression of genes involved in cell survival, metabolism, proliferation, and immune evasion (reviewed in ref. Dong *et al* (2019)), and proteins involved in the regulation of HSF1 have been reported to be involved in sensitivity to erastin-induced ferroptosis (Sun *et al*, 2015). Taken together, these data suggest a connection between HSF1, ferroptosis, and prominin2.

Initially, we evaluated the expression of HSF1 mRNA by qPCR in response to 4HNE or RSL3 and observed a significant increase that peaked at 15 min after treatment (Fig 3B). To ascertain the role of p38 in the increase in HSF1 expression, we pretreated MCF10A cells with the p38 inhibitor BIRB and then exposed the cells to 4HNE or RSL3. Inhibition of p38 reduced the 4HNE- or RSL3-driven increase in HSF1 to basal levels after 30 min of treatment (Fig 3C). HSF1 transcriptional activity can be regulated by its expression and phosphorylation on S326 (Guettouche *et al*, 2005). 4HNE and RSL3 increased HSF1 protein expression levels, as well as its phosphorylation at S326 after 60 min of exposure (Figs 3D and EV3A). We also observed that BIRB prevented the induction and phosphorylation of HSF1 following 4HNE or RSL3 treatment in both MCF10A and Hs578t cells, further indicating that this increase in HSF1 activity is dependent on p38 (Figs 3D and EV3A). Importantly, overexpression of prominin2 in MCF10A cells (Fig EV3C) diminished their sensitivity to p38 and HSF1 inhibition (Fig EV3D).

Nuclear localization is required for the transcriptional function of HSF1 (Sarge *et al*, 1993). Indeed, HSF1 expression and nuclear localization increased after 60 min of exposure to either 4HNE or RSL3 (Figs 3B and EV3B). Treatment with BIRB prior to 4HNE or RSL3 significantly reduced the increase in expression and nuclear localization of HSF1, substantiating the role of p38 in activating HSF1 (Fig 3D and E). These data describe a clear role for p38 in the increased expression and activation of HSF1 in response to lipid peroxidation.

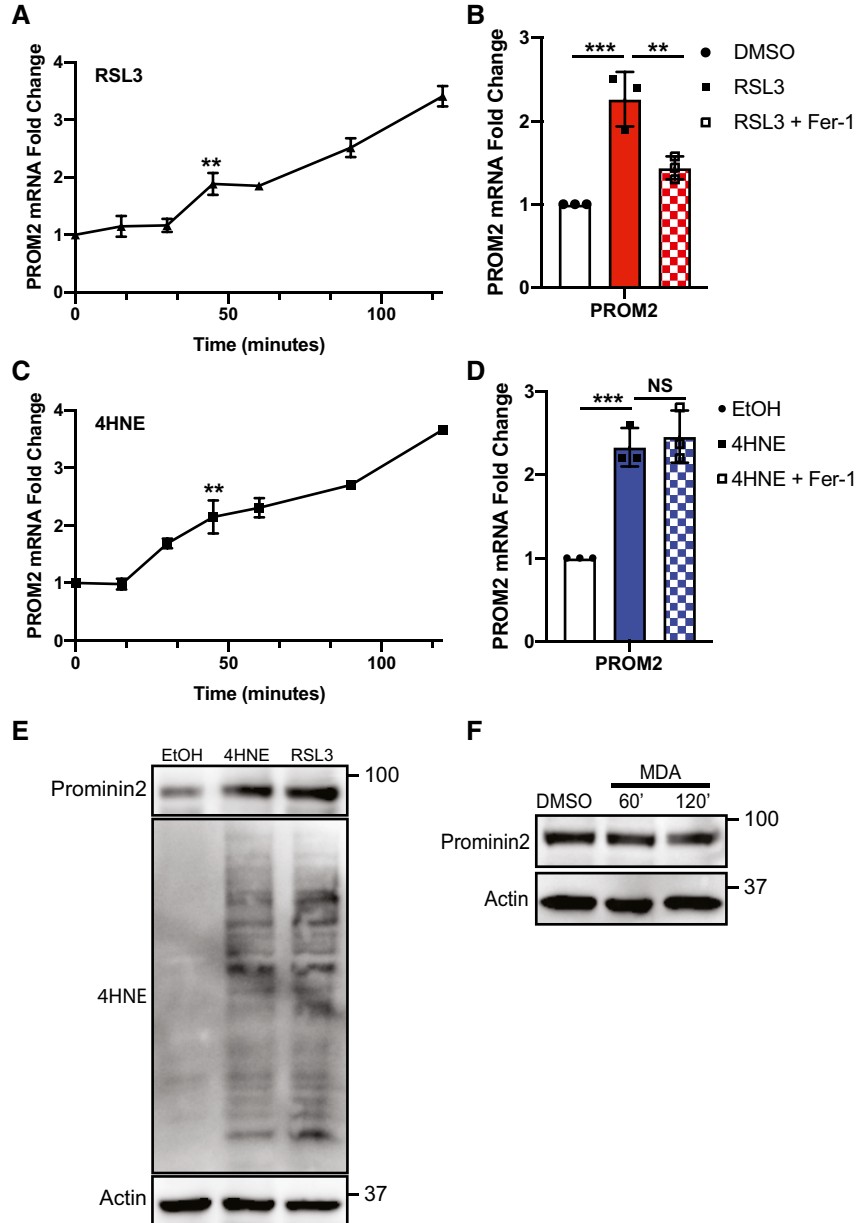

**Figure 1. RSL3 and the lipid peroxidation metabolite 4HNE trigger an increase in *PROM2* expression.**

A    MCF10A cells were treated with 5 μM RSL3 for times that ranged from 0 to 120 min. mRNA was isolated from each time point and *PROM2* expression was quantified by qPCR. Shown is an average of three independent experiments with standard deviation (*n* = 3 experiments per time point). *P*-values were obtained by unpaired Student's *t*-test with **P* < 0.05, ***P* < 0.01, ****P* < 0.005. Exact *P*-values are reported in Appendix Table S1.

B    MCF10A cells were pretreated for 15 min with either DMSO or 2 μM fer-1, followed by 60 min with either DMSO, 5 μM RSL3, or RSL3 and fer-1. mRNA was isolated and *PROM2* expression was quantified by qPCR. Shown are three independent experiments with standard deviation (*n* = 3 experiments per group). *P*-values were obtained by unpaired Student's *t*-test with **P* < 0.05, ***P* < 0.01, ****P* < 0.005. Exact *P*-values are reported in Appendix Table S1.

C    MCF10A cells were treated with 25 μM 4HNE for times that ranged from 0 to 120 min. mRNA was isolated from each time point, and *PROM2* expression was quantified by qPCR. Shown is an average of three independent experiments with standard deviation (*n* = 3 experiments per time point). *P*-values were obtained by unpaired Student's *t*-test with **P* < 0.05, ***P* < 0.01, ****P* < 0.005. Exact *P*-values are reported in Appendix Table S1.

D    MCF10A cells were pretreated for 15 min with either DMSO or 2 μM fer-1, followed by 60 min with either EtOH, 25 μM 4HNE, or RSL3 and fer-1. mRNA was isolated and *PROM2* expression was quantified by qPCR. Shown are three independent experiments with standard deviation (*n* = 3 experiments per group). *P*-values were obtained by unpaired Student's *t*-test with **P* < 0.05, ***P* < 0.01, ****P* < 0.005. *n* = 3 experiments per group. Exact *P*-values are reported in Appendix Table S1.

E    MCF10A cells were treated with either EtOH, 25 μM 4HNE, or 5 μM RSL3 for 60 min. Isolated protein was assessed by immunoblotting for 4HNE, prominin2, and β-actin expression. Shown is one replicate of three independent experiments (*n* = 3).

F    MCF10A cells were treated with DMSO or 100 μM MDA for 60 and 120 min. Isolated protein was assessed by immunoblotting for prominin2 and β-actin expression. Shown is one replicate of three independent experiments (*n* = 3).

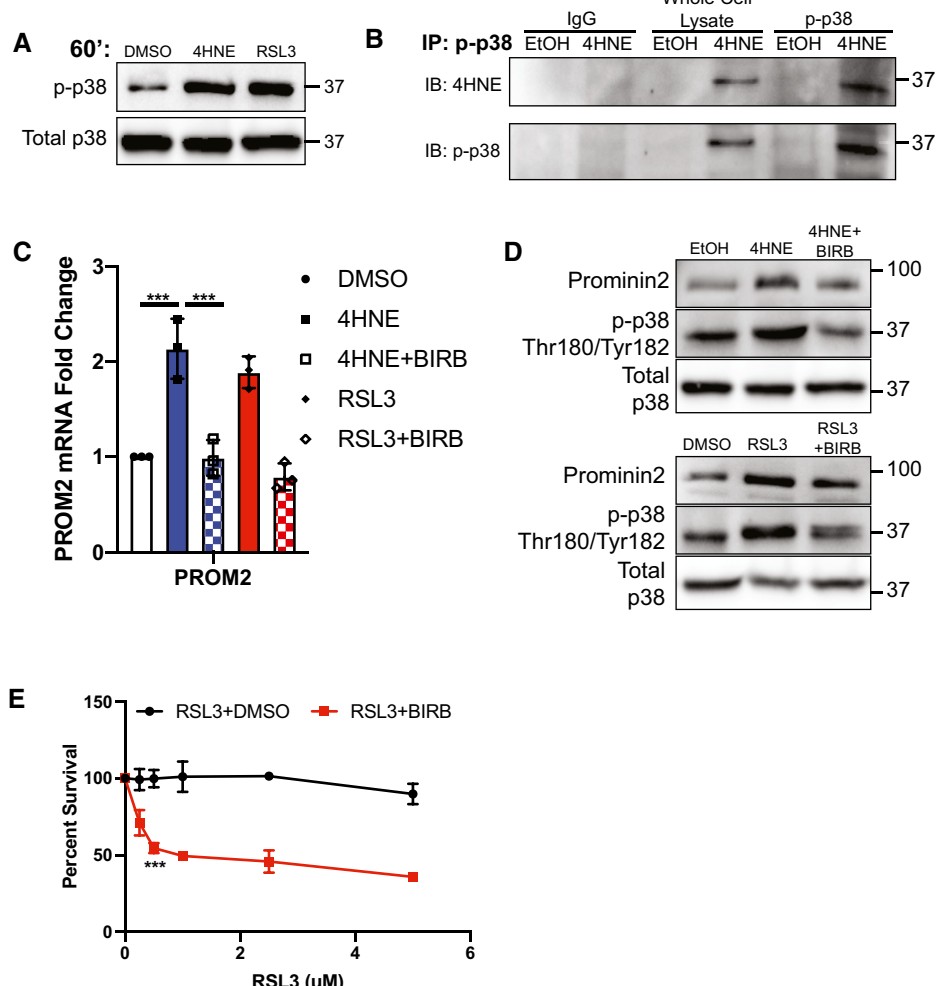

**Figure 2. 4HNE activates p38 to upregulate prominin2.**

A  MCF10A cells were treated with either DMSO, 25 μM 4HNE, or 5 μM RSL3 for 60 min. Isolated protein was assessed by immunoblotting for phospho-p38 (Thr180/Tyr182) and total p38 expression. Shown is one replicate of three independent experiments (*n* = 3).

B  MCF10A cells were treated with either EtOH or 25 μM 4HNE for 60 min. Isolated protein was assessed by immunoprecipitating phospho-p38 (Thr180/Tyr182) and immunoblotting for 4HNE and phospho-p38 (Thr180/Tyr182). Shown is one replicate of three independent experiments (*n* = 3).

C  MCF10A cells were pretreated for 15 min with either DMSO or 10 μM BIRB, followed by 60 min with either DMSO, 25 μM 4HNE, 5 μM RSL3, 4HNE, and BIRB or RSL3 and BIRB. mRNA was isolated and *PROM2* expression was quantified by qPCR. Shown are three independent experiments with standard deviation (*n* = 3 experiments per group). *P*-values were obtained by unpaired Student's *t*-test with \**P* < 0.05, \*\**P* < 0.01, \*\*\**P* < 0.005. Exact *P*-values are reported in Appendix Table S1.

D  MCF10A cells were pretreated for 15 min with either DMSO, EtOH, or 10 μM BIRB, followed by 60 min with either DMSO, EtOH, 25 μM 4HNE, 5 μM RSL3, 4HNE, and BIRB or RSL3 and BIRB. Isolated protein was assessed by immunoblotting for prominin2, phospho-p38 (Thr180/Tyr182), and total p38. Shown is one replicate of three independent experiments (*n* = 3).

E  MCF10A cells were treated with RSL3 with either DMSO or 10 μM BIRB. Cells were assessed for viability after 24 h. Absorbance was normalized to DMSO control. Shown are three independent replicates with standard deviation (*n* = 3 experiments per group). *P*-values were obtained by unpaired Student's *t*-test with \**P* < 0.05, \*\**P* < 0.01, \*\*\**P* < 0.005. Exact *P*-values are reported in Appendix Table S1.

Based on our finding that HSF1 is required for the increase in prominin2 expression, we investigated whether it is necessary for prominin2-mediated resistance to ferroptosis. MCF10A cells were transfected with pooled siRNAs (control or HSF1) as used in Ref. (Kammanadiminti & Chadee, 2006; Jeong *et al*, 2015) and assessed for their ability to increase prominin2 expression after either 4HNE or RSL3 treatment. Cells depleted of HSF1 were able to increase p38 activation in response to 4HNE but not to increase prominin2 expression compared to control cells (Fig 3F). Moreover, knockdown of HSF1 increased the sensitivity of MCF10A cells to RSL3, an

effect that was rescued partially by ferrostatin-1 (Figs 3G and EV3E) suggesting that non-ferroptotic cell death may also contribute. Taken together, these data implicate a key role for HSF1 in ferroptosis resistance by a mechanism that involves its ability to drive *PROM2* transcription.

### HSF1 inhibition sensitizes cells to ferroptotic stress

The data we obtained implicating HSF1 in ferroptosis resistance provided a potentially significant opportunity for sensitizing cells to

ferroptosis because specific HSF1 inhibitors are available. Because prominin2 inhibitors are not currently available, targeting HSF1 could be an effective bypass strategy. To address this possibility, we used KRIBB11, which directly targets HSF1 by inhibiting its ability to recruit the elongation factor pTEFb (Yoon *et al*, 2011). Inhibition of HSF1 activity by KRIBB11 prevented the upregulation of *PROM2* mRNA (Fig 4A) and prominin2 protein expression (Fig EV4A) in response to either 4HNE or RSL3. However, treatment with KRIBB11 alone had no significant effect on either *PROM2* or *HSF1* mRNA expression (Fig EV4 D and E). Moreover, neither the expression,

phosphorylation, nor nuclear localization of HSF1 was affected by KRIBB11 (Figs 4B, C, and D and EV4A), consistent with its reported mechanism of action (Yoon *et al*, 2011).

Having identified HSF1 as a key mediator of prominin2 expression and, consequently, the evasion of ferroptosis, we co-treated MCF10A cells with KRIBB11 and either RSL3, imidazole ketone erastin (IKE), or FIN56, compounds that induce ferroptosis through three separate mechanisms (Yang & Stockwell, 2008; Dixon *et al*, 2012; Yang *et al*, 2014; Shimada *et al*, 2016; Zhang *et al*, 2019). The ability of MCF10A cells to evade cell death with all three compounds was significantly

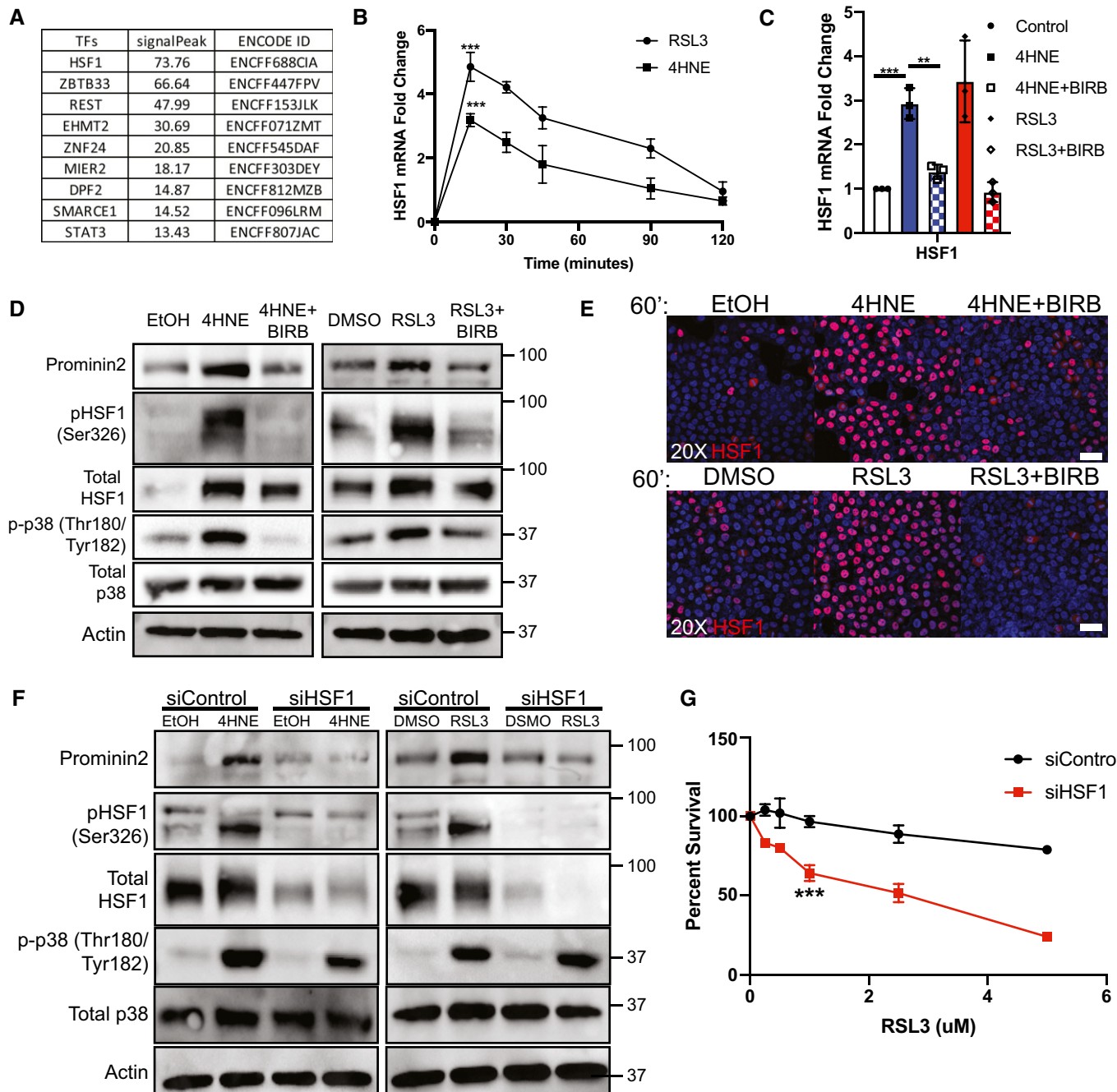

**Figure 3.**

**Figure 3. 4HNE activates HSF1 to induce prominin2 expression.**

A HSF1 was identified as a strong occupier of the *PROM2* promoter.
B MCF10A cells were treated with either 25 μM 4HNE or 5 μM RSL3 for times that ranged from 0 to 120 min. mRNA was isolated from each time point, and HSF1 expression was quantified by qPCR. Shown is an average of three independent experiments with standard deviation (*n* = 3 experiments per time point). *P*-values were obtained by unpaired Student's *t*-test with \**P* < 0.05, \*\**P* < 0.01, \*\*\**P* < 0.005. Exact *P*-values are reported in Appendix Table S1.
C MCF10A cells were pretreated for 15 min with either DMSO or 10 μM BIRB, followed by 60 min with either DMSO, 25 μM 4HNE, 5 μM RSL3, 4HNE, and BIRB or RSL3 and BIRB. mRNA was isolated and HSF1 expression was quantified by qPCR. Shown are three independent experiments with standard deviation (*n* = 3 experiments per group). *P*-values were obtained by unpaired Student's *t*-test with \**P* < 0.05, \*\**P* < 0.01, \*\*\**P* < 0.005. Exact *P*-values are reported in Appendix Table S1.
D MCF10A cells were pretreated for 15 min with either DMSO, EtOH, or 10 μM BIRB, followed by 60 min with either DMSO, EtOH, 25 μM 4HNE, 5 μM RSL3, 4HNE, and BIRB or RSL3 and BIRB. Isolated protein was assessed by immunoblotting for prominin2, phospho-HSF1 (S326), total HSF1, phospho-p38 (Thr180/Tyr182), total p38, and β-actin expression. Shown is one replicate of three independent experiments (*n* = 3).
E MCF10A cells were plated on slides pre-coated with laminin. Cells were then pretreated for 15 min with either DMSO, EtOH, or 10 μM BIRB, followed by 60 min with either DMSO, EtOH, 25 μM 4HNE, 5 μM RSL3, 4HNE, and BIRB or RSL3 and BIRB. Cells were stained for total HSF1 and counterstained with DAPI. Images were taken at 20x magnification. Scale bar is 50 μm. Shown is one replicate of three independent experiments (*n* = 3).
F HSF1 expression was diminished in MCF10A cells for 48 h using siRNA prior to treatment with either DMSO, EtOH, DMSO, 25 μM 4HNE, or 5 μM RSL3. Isolated protein was assessed by immunoblotting for prominin2, phospho-HSF1 (Ser326), total HSF1, phospho-p38 (Thr180/Tyr182), total p38, and β-actin expression. Shown is one replicate of three independent experiments (*n* = 3).
G HSF1 expression was diminished in MCF10A cells for 48 h using siRNA prior to treatment with either DMSO or RSL3 at the range of concentrations shown. Cells were assessed for viability after 24 h. Absorbance was normalized to DMSO control. Shown are three independent replicates with standard deviation (*n* = 3 experiments per group). *P*-values were obtained by unpaired Student's *t*-test with \**P* < 0.05, \*\**P* < 0.01, \*\*\**P* < 0.005. Exact *P*-values are reported in Appendix Table S1.

decreased when given in concert with KRIBB11 (Fig 4E). The cell death induced by the combination of IKE and KRIBB11 was rescued by co-treatment with ferrostatin-1 but not ZVAD-fmk, indicating that the cells were dying by ferroptosis (Fig EV4B). This phenomenon was specific to ferroptosis because co-treatment with other therapeutics including dasatinib and erlotinib did not significantly improve sensitivity (Fig EV4C). These data indicate that HSF1 specifically drives a ferroptosis-resistance program and that co-treatment with a ferroptosis-inducing agent and an HSF1 inhibitor can induce ferroptosis in otherwise resistant cells.

Although MCF10A cells, immortalized mammary epithelial cells, are considered "normal", their ability to survive in anoikis conditions suggests otherwise (Brown *et al*, 2017). To evaluate whether the combination of IKE and KRIBB11 would be toxic to more normal cells, we treated human mammary luminal epithelial cells (HMLE) and HMT-3522 S1 cells with these compounds. These cells are also immortalized but they approximate the normal cell state and are used as a point of comparison to transformed cells (Mani *et al*, 2008; Rizki

*et al*, 2008). Neither of these cell lines exhibited a notable decrease in viability after 24 h of treatment (Fig 4F and G). Similar results were obtained using RSL3 and KRIB11 (Fig EV4 F and G). These findings suggest that normal cells are relatively insensitive to this therapy, a conclusion substantiated by the *in vivo* data below.

We extended our analysis of the role of HSF1 in ferroptosis resistance to other cancer cell types. For this purpose, we used cell lines that are resistant to ferroptosis induced by GPX4 inhibition, including Hs578t cells, the glioblastoma cell line SF295, and the non-small cell lung cancer cell line NCI H1975. We assessed whether KRIBB11 increased their sensitivity to either RSL3 or IKE. Indeed, the ability to resist RSL3- or IKE-induced ferroptosis in all cell lines was significantly reduced when cells were co-treated with RSL3 and KRIBB11, although neither treatment alone had a significant effect (Fig 5A–C).

Although many cancer cells are highly sensitive to drugs that induce ferroptosis (Hangauer *et al*, 2017; Viswanathan *et al*, 2017), it is likely that such cells (and tumors) can acquire resistance to these drugs. To assess this possibility, we treated MDA-MB-231 cells

**Figure 4. HSF1 inhibition prevents prominin2-mediated evasion of ferroptosis.**

A MCF10A cells were treated for 60 min with either DMSO, 25 μM 4HNE, 5 μM RSL3, 4HNE, and 10 μM KRIBB11 or RSL3 and KRIBB11. mRNA was isolated, and *PROM2* expression was quantified by qPCR. Shown are three independent experiments with standard deviation (*n* = 3 experiments per group). *P*-values were obtained by unpaired Student's *t*-test with \**P* < 0.05, \*\**P* < 0.01, \*\*\**P* < 0.005. Exact *P*-values are reported in Appendix Table S1.
B MCF10A cells were treated for 60 min with either DMSO, 25 μM 4HNE, 5 μM RSL3, 4HNE, and 10 μM KRIBB11 or RSL3 and KRIBB11. mRNA was isolated and HSF1 expression was quantified by qPCR. Shown are three independent experiments with standard deviation (*n* = 3 experiments per group). *P*-values were obtained by unpaired Student's *t*-test with \**P* < 0.05, \*\**P* < 0.01, \*\*\**P* < 0.005. Exact *P*-values are reported in Appendix Table S1.
C MCF10A cells were plated on slides pre-coated with laminin. Cells were treated for 60 min with either EtOH, DMSO, 25 μM 4HNE, 5 μM RSL3, 4HNE and 10 μM KRIBB1 or RSL3 and KRIBB11. Cells were stained for total HSF1 and counterstained with DAPI. Images were taken at 20x magnification. Scale bar is 50 μm. Shown is one replicate of three independent experiments (*n* = 3).
D MCF10A cells were treated for 60 min with either EtOH, DMSO, 25 μM 4HNE, 5 μM RSL3, 4HNE, and 10 μM KRIBB11 or RSL3 and KRIBB11. Isolated protein was assessed by immunoblotting for prominin2, phospho-HSF1 (Ser326), total HSF1, phospho-p38 (Thr180/Tyr182), total p38, and β-actin expression. Shown is one replicate of three independent experiments (*n* = 3).
E MCF10A cells were treated with either RSL3, FIN56, or IKE at the range of concentrations shown with either DMSO or 10 μM KRIBB11. Cells were assessed for viability after 24 h. Absorbance was normalized to DMSO control. Shown are three independent replicates with standard deviation (*n* = 3 experiments per group). *P*-values were obtained by unpaired Student's *t*-test with \**P* < 0.05, \*\**P* < 0.01, \*\*\**P* < 0.005. Exact *P*-values are reported in Appendix Table S1.
F HMLE cells were treated with IKE at the range of concentrations shown with either DMSO or 10 μM KRIBB11. Cells were assessed for viability after 24 h. Absorbance was normalized to DMSO control. Shown are three independent replicates with standard deviation (*n* = 3 experiments per group). *P*-values were obtained by unpaired Student's *t*-test with \**P* < 0.05, \*\**P* < 0.01, \*\*\**P* < 0.005. Exact *P*-values are reported in Appendix Table S1.
G S1 cells were treated with IKE at the range of concentrations shown with either DMSO or 10 μM KRIBB11. Cells were assessed for viability after 24 h. Absorbance was normalized to DMSO control. Shown are three independent replicates with standard deviation (*n* = 3 experiments per group). *P*-values were obtained by unpaired Student's *t*-test with \**P* < 0.05, \*\**P* < 0.01, \*\*\**P* < 0.005. Exact *P*-values are reported in Appendix Table S1.

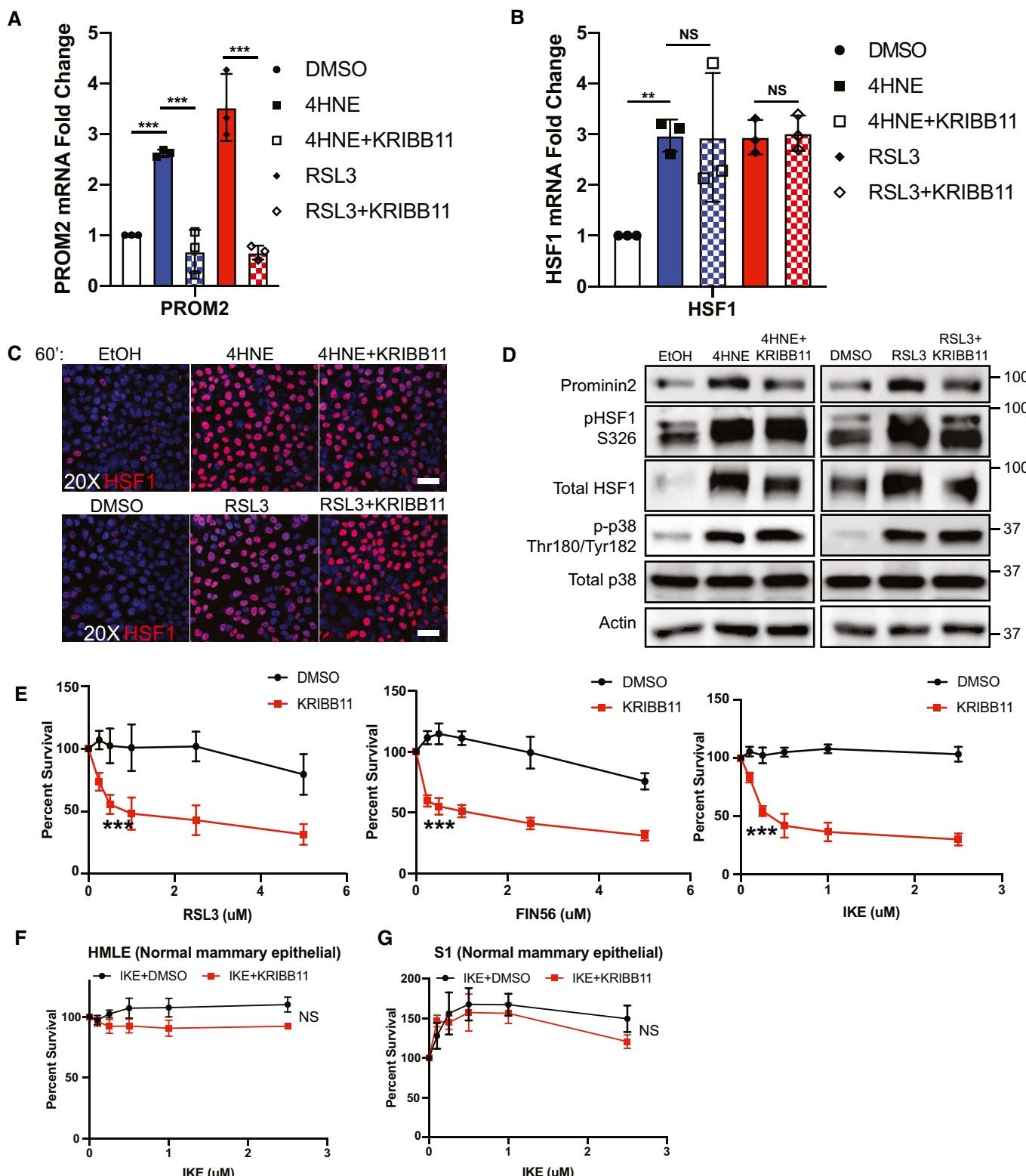

**Figure 4.**

with RSL3 for several weeks and were able to select for a population of cells that resisted this treatment. We then evaluated the ability of the parental, sensitive cells (MDA-MB-231S) and resistant (MDA-MB-231R) cells to survive treatment with RSL3 or IKE singly or in combination with KRIBB11 (Figs 5D and E, and EV5A and B). MDA-MB-231S cells were unable to resist ferroptosis induced by IKE, either with or without KRIBB11, at concentrations as low as 0.5 μM. In contrast, MDA-MB-231R cells were able to survive in

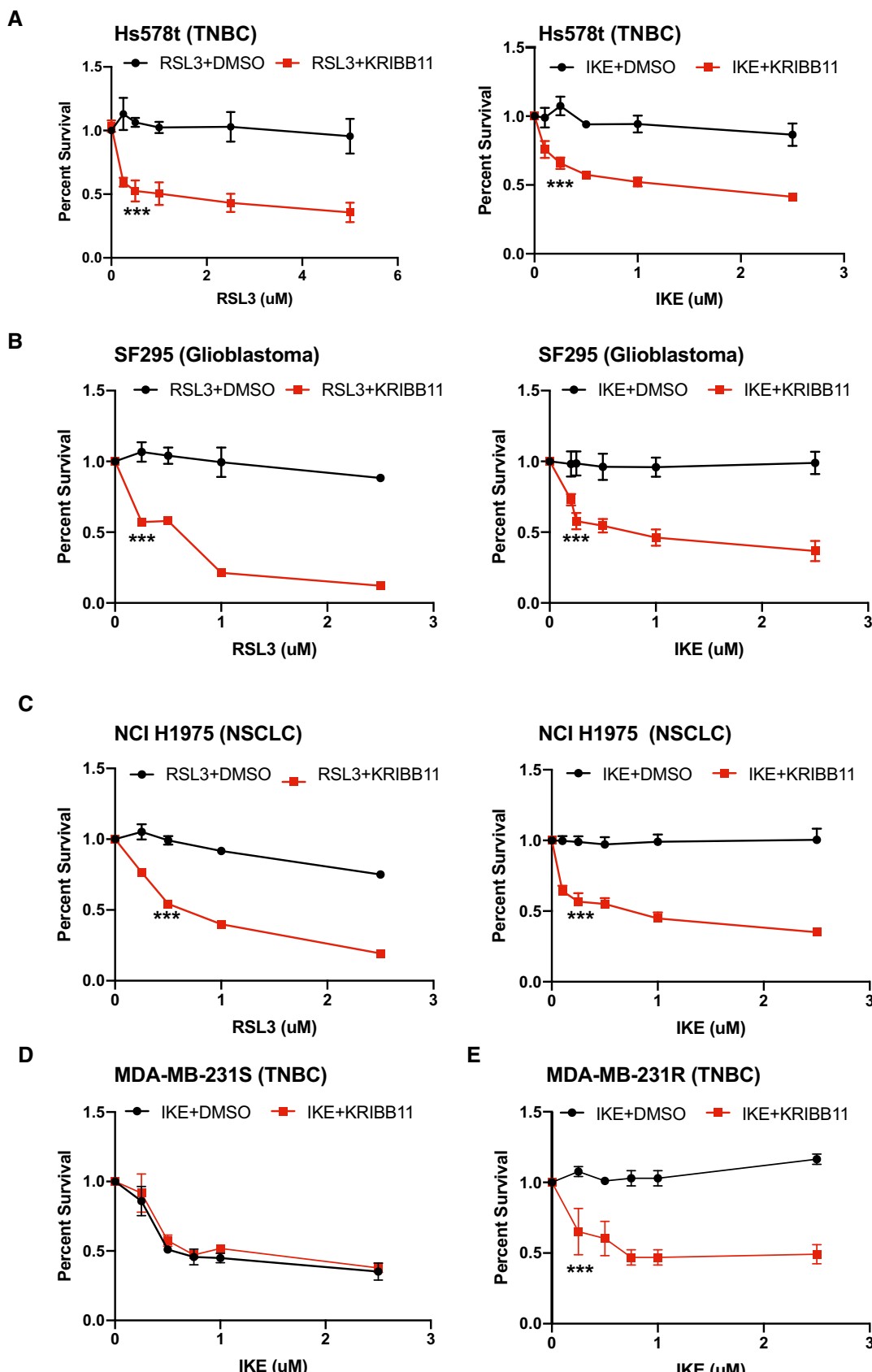

Figure 5.

**Figure 5. HSF1 inhibition sensitizes cancer cells to ferroptotic cell death.**

A   Hs578t cells were treated with either RSL3 or IKE at the range of concentrations shown in combination with either DMSO or 10 µM KRIBB11. Cells were assessed for viability after 24 h. Absorbance was normalized to DMSO control. Shown are three independent replicates with standard deviation ($n = 3$ experiments per group). $P$-values were obtained by unpaired Student's $t$-test with $*P < 0.05$, $**P < 0.01$, $***P < 0.005$. Exact $P$-values are reported in Appendix Table S1.
B   SF295 cells were treated with either RSL3 or IKE at the range of concentrations shown in combination with either DMSO or 10 µM KRIBB11. Cells were assessed for viability after 24 h. Absorbance was normalized to DMSO control. Shown are three independent replicates with standard deviation ($n = 3$ experiments per group). $P$-values were obtained by unpaired Student's $t$-test with $*P < 0.05$, $**P < 0.01$, $***P < 0.005$. Exact $P$-values are reported in Appendix Table S1.
C   NCI H1975 cells were treated with either RSL3 or IKE at the range of concentrations shown in combination with either DMSO or 10 µM KRIBB11. Cells were assessed for viability after 24 h. Absorbance was normalized to DMSO control. Shown are three independent replicates with standard deviation ($n = 3$ experiments per group). $P$-values were obtained by unpaired Student's $t$-test with $*P < 0.05$, $**P < 0.01$, $***P < 0.005$. Exact $P$-values are reported in Appendix Table S1.
D   MDA-MB-231 S cells were treated with IKE at the range of concentrations shown in combination with either DMSO or 10 µM KRIBB11. Cells were assessed for viability after 24 h. Absorbance was normalized to DMSO control. Shown are three independent replicates with standard deviation ($n = 3$ experiments per group). $P$-values were obtained by unpaired Student's $t$-test with $*P < 0.05$, $**P < 0.01$, $***P < 0.005$. Exact $P$-values are reported in Appendix Table S1.
E   MDA-MB-231R cells were treated with IKE at the range of concentrations shown in combination with either DMSO or 10 µM KRIBB11. Cells were assessed for viability after 24 h. Absorbance was normalized to DMSO control. Shown are three independent replicates with standard deviation ($n = 3$ experiments per group). $P$-values were obtained by unpaired Student's $t$-test with $*P < 0.05$, $**P < 0.01$, $***P < 0.005$. Exact $P$-values are reported in Appendix Table S1.

concentrations of IKE up to 2.5 µM. However, the combination of IKE with KRIBB11 was sufficient to overcome the resistance of the MDA-MB-231R cells to ferroptosis, consistent with the mechanism we have described.

## HSF1 inhibition sensitizes IKE-resistance cells *in vivo*

The combination of IKE and KRIBB11 caused a significant decrease in the survival of several cancer cell lines compared with either compound alone (Fig 5A–C). Although these *in vitro* data were compelling, it was important to assess the ability of this combinatorial approach to impede tumor growth *in vivo*. For this purpose, we used Hs578t cells because they are resistant to ferroptosis (Brown *et al*, 2019) including treatment with IKE (Fig 5A). We implanted Hs578t cells in the mammary fat pads of NOD/SCID mice and monitored tumor formation over 21 days (Fig 6A). Mice were randomized into treatment groups of vehicle (5% DMSO, 95% HBSS), IKE alone (23 mg/kg), or a combination of IKE and KRIBB11 (50 mg/kg) with daily intraperitoneal injections. During the treatment period, mice were weighed and assessed for tumor burden daily. Administration of IKE + KRIBB11 caused a significant decrease in tumor growth compared with either vehicle alone or IKE (Fig 6B). IKE treatment alone did not impede tumor growth consistent with our finding that Hs578t cells are resistant to ferroptosis (Fig 5A) (Brown *et al*, 2019). Importantly, the combination of therapeutics did not adversely affect mouse weight, suggesting a low level of toxicity (Fig 6C), in agreement with our observation that normal mammary epithelial cells were not affected by combined exposure to IKE and KRIBB11 *in vitro* (Fig 4F).

Evaluation of tumors by immunocytochemistry showed a strong induction of prominin2 in the IKE-treated tumors that was absent in tumors treated with IKE + KRIBB11 (Fig 6D). While the increase in prominin2 staining was absent in IKE + KRIBB11 tumors, HSF1 staining increased in both IKE- and IKE + KRIBB11-treated tumors (Fig 6D), in agreement with the mechanism of action of prominin2 induction and ferroptosis resistance. Notably, 4HNE staining increased with IKE treatment but decreased with the combined treatment compared to the control, supporting the conclusion that IKE + KRIBB11 increases lipid peroxidation while KRIBB11 itself prevents the induction of prominin2.

## Discussion

The data we present here provide a novel strategy for inducing ferroptosis in cancer cells that are otherwise resistant to this mode of cell death. We were able to exploit a finding that prominin2 has a key role in ferroptosis resistance by uncovering the mechanism by which it is induced in response to ferroptotic stress and develop a strategy to target this mechanism (Fig 7). As a result, we found that the combined use of drugs that have the potential to induce ferroptosis with a drug that inhibits *PROM2* transcription causes resistant cells to succumb to ferroptosis.

We discovered that ferroptosis resistance is triggered by 4HNE, a lipid metabolite that is formed as an initial response to inhibition of either GPX4 or xCT. Interestingly, other lipid peroxidation products including MDA, 4ONE, and 4HHE were unable to induce prominin2 expression. Although this finding argues for the specificity of 4HNE in regulating prominin2, we cannot exclude the contribution of other lipid peroxidation products that were not evaluated. Nonetheless, the ability of resistant cells to activate p38 MAPK in response to 4HNE and promote activation of HSF1 that induces prominin2 expression indicates that cancer cells can exhibit a coordinated signaling response that protects them from ferroptosis.

**Figure 6. Inhibition of HSF1 increases sensitivity to IKE *in vivo*.**

A   Schematic of experimental design. Figure partially created using BioRender.
B   Tumor volume was measured daily by electronic caliper. Data are plotted as the mean ± SD for $n = 8$ individual mice per group. $P$-values were obtained by two-way ANOVA with $*P < 0.05$, $**P < 0.01$, $***P < 0.005$. Exact $P$-values are reported in Appendix Table S1.
C   Mouse weight was monitored daily and indicated no significant weight loss with IKE + KRIBB11. Data are plotted as the mean ± SD for $n = 8$ individual mice per group.
D   Twenty-four hours after the last treatment, tumors were excised, frozen, and sectioned. Each tumor was stained with Abs specific for either 4HNE, prominin2, or HSF1. Images were taken at 20x magnification. Higher magnification images (1.5×) are shown to the right of its corresponding inset (white box). Scale bar is 50 µm. Shown is one representative image for each condition.

**A**

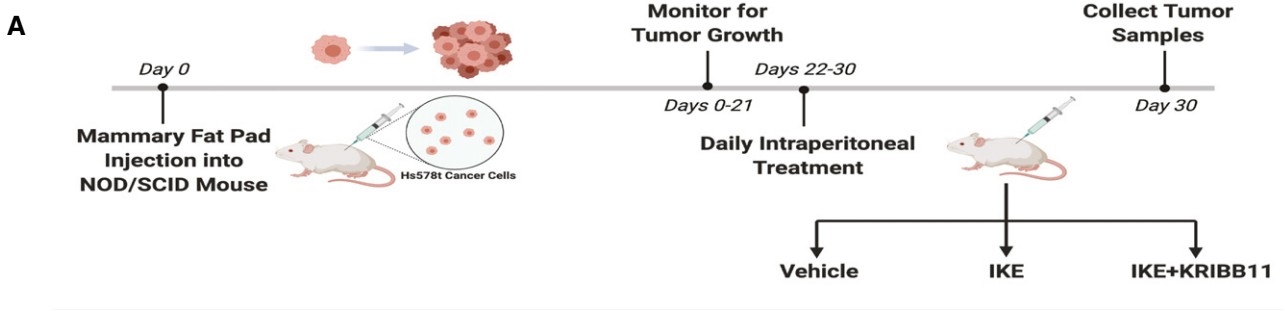

**B** 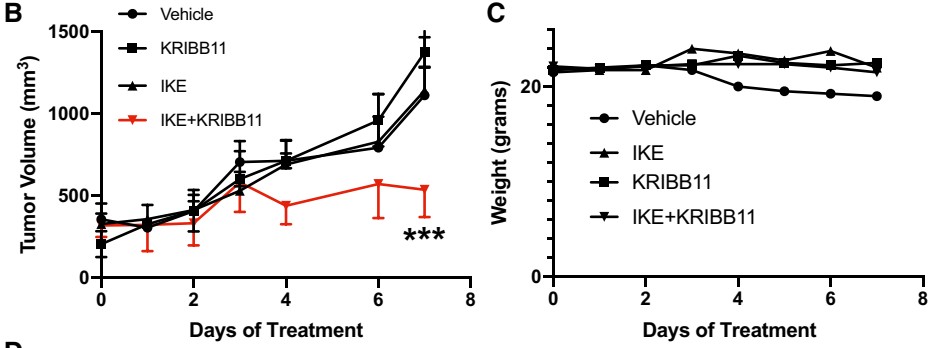 **C**

**D**

Figure 6.

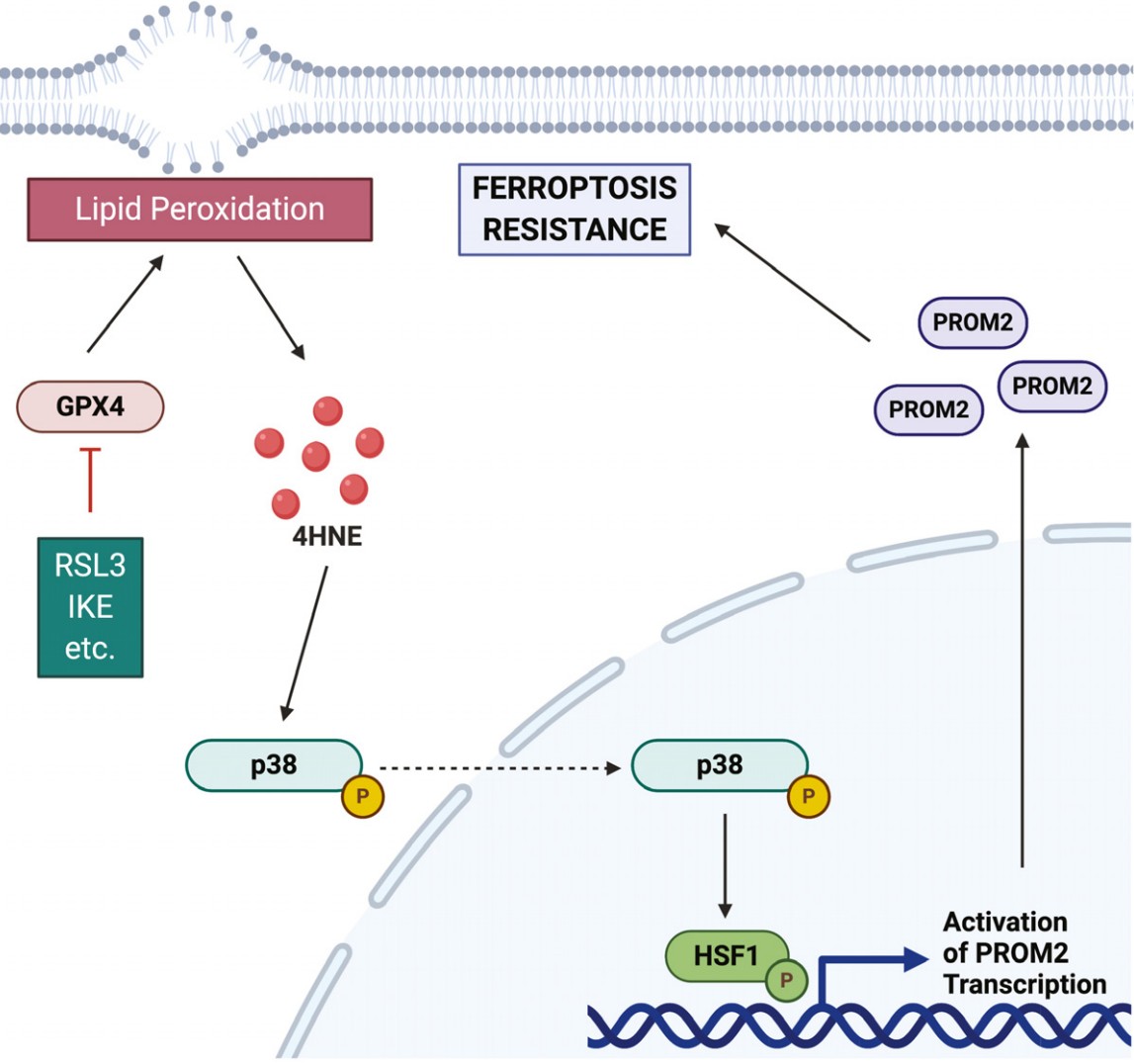

**Figure 7. Schematic of the proposed mechanism for prominin2 induction mechanism by lipid peroxidation.**

The lipid peroxidation metabolite 4HNE activates p38/MAPK to phosphorylate and activate the stress-responsive transcription factor HSF1. The consequent increase in prominin2 expression promotes ferroptosis resistance, and this mechanism can be targeted to enable resistant cells to succumb to ferroptosis. Figure partially created using BioRender.

In recent years, there has been intense interest in developing drugs and approaches that stimulate ferroptosis in cancer as a therapeutic strategy. A key finding in this direction is that cancer cells with a more de-differentiated, mesenchymal phenotype are dependent on GPX4 for their survival and, consequently, more sensitive to ferroptosis triggered by GPX4 inhibition than their more differentiated counterparts (Hangauer *et al*, 2017; Viswanathan *et al*, 2017). Nonetheless, many cancer cells including aggressive breast, glioblastoma, and lung cancer resist ferroptosis as we show in Fig 5A–C. Moreover, cells that are sensitive to GPX4 inhibition can acquire resistance and this resistance, as we have shown here, that can be undermined by inhibiting both GPX4 and HSF1 (Fig 5D and E). For these reasons, our findings could have a significant impact on the approach to induction of ferroptosis as a therapeutic strategy in cancer. This conclusion is supported by our finding that tumor xenografts that are resistant to IKE were sensitized to this

compound by combined treatment with KRIBB11, presumably because inhibition of HSF1 and the consequent reduction in prominin2 expression enabled IKE to induce ferroptosis. It should be noted, however, that the ability of IKE to induce ferroptosis *in vivo* has not been unequivocally demonstrated as it has *in vitro*. Nonetheless, the data we present validate our hypothesis that targeting mechanisms that promote resistance to drugs, which have the potential to induce ferroptosis, is a viable therapeutic strategy.

# Materials and Methods

### Cell lines and reagents

MCF10A cells were obtained from the Barbara Ann Karmanos Cancer Institute. Hs578t cells were provided by D. Kim (University

of Massachusetts Medical School, Worcester, MA), SW640, SW860, NCI H1975, and SF295 cells were provided by M. Green (University of Massachusetts Medical School, Worcester, MA), HMLE cells were provided by R. Weinberg (Massachusetts Institute of Technology, Cambridge, MA), and S1 cells were provided by M. Bissell (Lawrence Berkeley National Laboratory, Berkeley, CA). The MCF10A cell line was maintained in DMEM/F12 containing 5% horse serum and 1% penicillin/streptomycin supplemented with 20 ng/ml epidermal growth factor (EGF), 0.5 mg/ml hydrocortisone, 100 ng/ml cholera toxin, and 10 μg/ml insulin. The Hs578t cell line was maintained in high-glucose DMEM (4,500 mg/l glucose) containing 5% fetal bovine serum, 1% penicillin/streptomycin, and 1% insulin. The SW640 and SW860 cell lines were maintained in Leibovitz's L-15 containing 10% fetal bovine serum and 1% penicillin/streptomycin. The NCI H1975 and SF 295 cell lines were maintained in RPMI-1640 containing 10% fetal bovine serum and 1% penicillin/streptomycin.

The S1 cell line was maintained in DMEM/F12 containing 250 ng/ml insulin, 10 μg/ml transferrin, 2.6 ng/ml sodium selenite, $10^{-10}$ M estradiol, 1.4 μM hydrocortisone, 5 μg/ml prolactin, and 10 ng/ml EGF. The HMLE cell line was maintained using MEGM Mammary Epithelial Cell Growth Medium BulletKit (Lonza).

All cell lines were regularly tested for possible mycoplasma contamination. All cell lines used in this study were mycoplasma negative.

The following antibodies were used: Prominin2 (Abcam), p-p38, total p38, total HSF1, actin, (Cell Signaling Technology), 4HNE (Abcam and Thermo Fisher), pHSF1 (Thermo Fisher), and Tubulin (Sigma-Aldrich). Other reagents used were: 4HNE (Cayman Chemical), RSL3, erastin, FIN56, ZVAD-fmk (Selleckchem), IKE (Med Chem Express), MDA (Sigma-Aldrich), KRIBB11 (Tocris, Med Chem Express), ferric ammonium citrate, ferrostatin-1 (Sigma-Aldrich).

### RNA interference

siRNA-mediated silencing of gene expression of HSF1 was performed with Lipofectamine 3000 per the manufacturer's guidelines using pooled control or HSF1 siRNAs (Santa Cruz Biotechnology). Cells were transfected and after 48 h were plated for experimental use.

### Cell survival assay

Cells were seeded in a 96-well plate to 50% confluence. Cells were treated with the compounds shown at the concentrations described. Twenty-four hours later, cells were assessed for viability by fixation in 4% paraformaldehyde (Boston BioProducts) for 15 min and then stained with crystal violet for 60 min. Plates were washed and dried before reading by absorbance at 595. Percent survival was calculated by comparing to the vehicle control.

### Biochemical experiments

For immunoblotting, cell lysates were extracted using radioimmunoprecipitation assay (RIPA) buffer (Boston BioProducts). Extracts were separated by SDS–PAGE using the specified antibodies. Primary antibodies for immunoblotting were diluted in 5% BSA in PBST at 1:1,000. Secondary antibodies for immunoblotting were diluted in 5% BSA in PBST at 1:5,000. Immunoprecipitation was performed on lysates isolated using NP-40 buffer (Boston BioProducts). Lysates

were pre-cleared prior to overnight incubation with the primary antibody and protein A agarose beads (Sigma-Aldrich). Immune complexes were separated by SDS–PAGE and immunoblotted as described in figure legends. RNA was isolated from cells (Bio Basic), and cDNA was generated using the All-in-One cDNA Synthesis SuperMix (BioScript Solutions). qPCR was performed using a SYBR green master mix (BioScript Solutions). Sequences for primers used were as follows: 18s forward 5'-AACCCGTTGA ACCCCATT-3', reverse 5'-CCATCCAATCG GTAGTAGCG-3', *PROM2* forward 5'- GCTC AGGAACC CAAACCTGT-3', reverse 5'- GGCAGGCCATAC ATCCTTC T-3', *HSF1* forward 5'-GAAGCAGCTGGTGCACTACA-3' reverse 5'-AAGTAGGAGCCCTCTCCCAG-3'.

### Immunofluorescence microscopy

Eight-well chamber slides (Mattek) were coated with 1 mg/ml laminin. Cells ($5.0 \times 10^4$ per well) were plated overnight and then treated for 1 h with either DMSO or RSL3, fixed in 4% paraformaldehyde (Boston BioProducts), and permeabilized in 0.1% Triton. Slides were blocked in 0.5% BSA, incubated in antibody overnight at 4°C, washed in PBS, and incubated in secondary antibody for 1 h at room temperature. Primary antibodies for immunofluorescence were diluted in 0.1% BSA in PBST at 1:500. Secondary antibodies for immunofluorescence were diluted in 0.1% BSA in PBST at 1:2,500. Slides were washed and mounted in Vectashield Anti-fade with DAPI. Slides were imaged on a Zeiss confocal microscope.

Tumor sections were fixed and permeabilized in ice-cold acetone for 10 min, washed 3× in PBS, and blocked in 1% BSA, 22.52 mg/ml glycine in PBST for 30 min. Sections were incubated overnight at 4°C in primary antibody at 1:500, then washed 3× in PBS and incubated with secondary at 1:2,500 for one hour at room temperature. Sections were washed 3× in PBS and mounted in Vectashield with DAPI, dried, and then imaged on a Zeiss confocal microscope.

### MAGIC matrix search

A previous study provided a compilation of multiple NarrowPeak files from ENCODE ChIP-seq experiments where they extracted the maximum peak signal (signalValue) found within each gene region (https://doi.org/10.1371/journal.pcbi.1007800). The gene region in this case is defined as the gene body, the promoter region, and a 5-kb flanking region on both sides of the gene in order to identify possible enhancers. We sorted for the top 10 *PROM2* interacting transcription factors as defined by the highest peak identified from a ChIP experiment for this transcription factor.

### *In vivo* experiments

All animal study protocols (UMMS Protocol ID: LEGACY1751-18 and PROTO202000145) were approved by the University of Massachusetts Medical School Institutional Animal Care and Use Committee (IACUC). 24 NOD/SCID mice (8 weeks of age) were weighed and injected in the mammary fat pad with Hs578t cells ($10^6$). Tumors were observable by palpation 21 days after injection. At 100 mm³, mice (8 mice/group) were separated randomly into groups for daily intraperitoneal injection of vehicle control (5% DMSO/95% HBSS), 50 mg/kg KRIBB11, 23 mg/kg IKE, or

**The paper explained**

**Problem**

The ability of tumor cells to resist cytotoxic therapies is a major challenge for many cancers, especially more aggressive tumors. This problem is particularly apt for recent therapeutic approaches aimed at inducing ferroptosis, a regulated form of non-apoptotic cell death that is iron-dependent and characterized by the accumulation of peroxidated lipids. Although some tumor cells are sensitive to drugs that trigger ferroptosis, others are not and determining the mechanism of this resistance is needed to improve the efficacy of this therapeutic approach. We had reported that cells that are resistant to ferroptosis are characterized by their ability to induce expression of prominin2 that functions to transport iron out of the cell and, consequently, preventing ferroptotic death. Although specific prominin2 inhibitors are not available, we rationalized that understanding the mechanism by which prominin2 expression is induced by ferroptotic stress could reveal other therapeutic targets.

**Results**

We discovered that the transcription of PROM2 is regulated by heat shock factor 1 (HSF1). HSF1 inhibitors are available, and we observed that they sensitized a wide variety of resistant cancer cells to drugs that induce ferroptosis. Also, the combination of a ferroptosis-inducing drug and an HSF1 inhibitor caused the cytostasis of established tumors in mice, although neither treatment alone was effective.

**Impact**

Our findings could have a significant impact on the approach to induction of ferroptosis as a therapeutic strategy in cancer including the treatment of tumors that have acquired resistance to drugs that trigger ferroptosis.

IKE+KRIBB11. All solutions were sterilized by filtering through a 0.2 μM syringe filter. Tumors were measured daily with electronic calipers, and volume was calculated by length (mm) × width (mm) × width (mm). Mice were sacrificed by $CO_2$ inhalation when tumor burden was more than 1,000 mm$^3$ or percent weight loss was above 15%. Tumors were collected and frozen for immunocytochemistry. Tumor volume changes were analyzed in GraphPad Prism using two-way ANOVA. All mice were housed under specific pathogen-free conditions maintained in an accredited animal facility at UMass Medical School. Mice were housed with a 12-h day–night cycle with lights on at 07:00AM in a temperature (22 ± 1°C)- and humidity (55 ± 5%)-controlled room. All mice were allowed free access to water and food. All cages contained paper nesting material, bedding, and cardboard for environmental enrichment. All mice were obtained from Jackson Laboratory (Bar Harbor, ME USA) and were acclimatized for at least 1 week prior to mammary fat pad injection. Vendor health reports indicated that the mice were free of known viral, bacterial, and parasitic pathogens.

**Statistical analysis**

All experiments were independently repeated at least three times. Statistical analysis was performed using GraphPad Prism software version 8.2. The *P*-value was calculated using Student's *t*-test or two-way ANOVA, and a $P < 0.05$ was considered significant. The bars in graphs represent mean ± SEM. $*P < 0.05$, $**P < 0.01$, $***P < 0.005$.

## Data availability

This study contains no data deposited in external repositories.

**Expanded View** for this article is available online.

## Acknowledgments
This work was supported by NIH grant CA218085 (A.M.M.) and ACS grant 130451-PF-17-105-01-CSM (C.W.B.).

## Author contributions
CWB, PC, and AMM designed the experiments and analyzed data. CWB, PC, and DM performed experiments. ERK performed the bioinformatics analysis. CWB, PC, and AMM wrote the manuscript.

## Conflict of interest
The authors declare that they have no conflict of interest.

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
