## [Review Process File · EMBO Molecular Medicine]

Targeting prominin2 transcription to overcome ferroptosis resistance in cancer

Caitlin Brown, Peter Chhoy, Dimpi Mukhopadhyay, Emmet Karner, and Arthur Mercurio

DOI: [10.15252/emmm.202013792](https://doi.org/10.15252/emmm.202013792)

Corresponding author(s): Arthur Mercurio (arthur.mercurio@umassmed.edu)

Review Timeline:

Submission Date:	2nd Dec 20
Editorial Decision:	7th Jan 21
Revision Received:	10th Apr 21
Editorial Decision:	22nd Apr 21
Author correspondence:	5 May 21
Editor correspondence:	6 May 21
Revision Received:	14th May 21
Editorial Decision:	21st May 21
Revision Received:	3rd Jun 21
Accepted:	7th Jun 21

Editor: Lise Roth

Transaction Report:

7th Jan 2021

Dear Dr. Mercurio,

Thank you for the submission of your manuscript to EMBO Molecular Medicine. We have now received feedback from the three reviewers who agreed to evaluate your manuscript. As you will see from the reports below, the referees acknowledge the interest of the study and are overall supporting publication of your work pending appropriate revisions.

Addressing the reviewers' concerns in full will be necessary for further considering the manuscript in our journal, and acceptance of the manuscript will entail a second round of review. EMBO Molecular Medicine encourages a single round of revision only and therefore, acceptance or rejection of the manuscript will depend on the completeness of your responses included in the next, final version of the manuscript. For this reason, and to save you from any frustrations in the end, I would strongly advise against returning an incomplete revision.

When submitting your revised manuscript, please carefully review the instructions that follow below. Failure to include requested items will delay the evaluation of your revision:

- 1) A .docx formatted version of the manuscript text (including legends for main figures, EV figures and tables). Please make sure that the changes are highlighted to be clearly visible.
- 2) Individual production quality figure files as .eps, .tif, .jpg (one file per figure).
- 3) A .docx formatted letter INCLUDING the reviewers' reports and your detailed point-by-point responses to their comments. As part of the EMBO Press transparent editorial process, the point-by-point response is part of the Review Process File (RPF), which will be published alongside your paper.
- 4) A complete author checklist, which you can download from our author guidelines (<https://www.embopress.org/page/journal/17574684/authorguide#submissionofrevisions>). Please insert information in the checklist that is also reflected in the manuscript. The completed author checklist will also be part of the RPF.
- 5) Please note that all corresponding authors are required to supply an ORCID ID for their name upon submission of a revised manuscript.
- 6) Before submitting your revision, primary datasets produced in this study need to be deposited in an appropriate public database (see <https://www.embopress.org/page/journal/17574684/authorguide#dataavailability>). Please remember to provide a reviewer password if the datasets are not yet public. The accession numbers and database should be listed in a formal "Data Availability" section (placed after Materials & Method). Please note that the Data Availability Section is restricted to new primary data that are part of this study.

7) We would also encourage you to include the source data for figure panels that show essential data. Numerical data should be provided as individual .xls or .csv files (including a tab describing the data). For blots or microscopy, uncropped images should be submitted (using a zip archive if multiple images need to be supplied for one panel). Additional information on source data and instruction on how to label the files are available at

8) Our journal encourages inclusion of *data citations in the reference list* to directly cite datasets that were re-used and obtained from public databases. Data citations in the article text are distinct from normal bibliographical citations and should directly link to the database records from which the data can be accessed. In the main text, data citations are formatted as follows: "Data ref: Smith et al, 2001" or "Data ref: NCBI Sequence Read Archive PRJNA342805, 2017". In the Reference list, data citations must be labeled with "[DATASET]". A data reference must provide the database name, accession number/identifiers and a resolvable link to the landing page from which the data can be accessed at the end of the reference. Further instructions are available at .

9) We replaced Supplementary Information with Expanded View (EV) Figures and Tables that are collapsible/expandable online. A maximum of 5 EV Figures can be typeset. EV Figures should be cited as 'Figure EV1, Figure EV2' etc... in the text and their respective legends should be included in the main text after the legends of regular figures.

- Additional Tables/Datasets should be labeled and referred to as Table EV1, Dataset EV1, etc. Legends have to be provided in a separate tab in case of .xls files. Alternatively, the legend can be supplied as a separate text file (README) and zipped together with the Table/Dataset file. See detailed instructions here:

10) The paper explained: EMBO Molecular Medicine articles are accompanied by a summary of the articles to emphasize the major findings in the paper and their medical implications for the non-specialist reader. Please provide a draft summary of your article highlighting

11) For more information: There is space at the end of each article to list relevant web links for further consultation by our readers. Could you identify some relevant ones and provide such information as well? Some examples are patient associations, relevant databases, OMIM/proteins/genes links, author's websites, etc...

12) Author contributions: the contribution of every author must be detailed in a separate section

(before the acknowledgments).

13) A Conflict of Interest statement should be provided in the main text

14) Every published paper now includes a 'Synopsis' to further enhance discoverability. Synopses are displayed on the journal webpage and are freely accessible to all readers. They include a short stand first (maximum of 300 characters, including space) as well as 2-5 one-sentences bullet points that summarizes the paper. Please write the bullet points to summarize the key NEW findings. They should be designed to be complementary to the abstract - i.e. not repeat the same text. We encourage inclusion of key acronyms and quantitative information (maximum of 30 words / bullet point). Please use the passive voice. Please attach these in a separate file or send them by email, we will incorporate them accordingly.

Please also suggest a striking image or visual abstract to illustrate your article as a png file 550 px-wide x 400-px high.

15) As part of the EMBO Publications transparent editorial process initiative (see our Editorial at <http://embomolmed.embopress.org/content/2/9/329>), EMBO Molecular Medicine will publish online a Review Process File (RPF) to accompany accepted manuscripts.

In the event of acceptance, this file will be published in conjunction with your paper and will include the anonymous referee reports, your point-by-point response and all pertinent correspondence relating to the manuscript. Let us know whether you agree with the publication of the RPF and as here, if you want to remove or not any figures from it prior to publication.

I look forward to receiving your revised manuscript.

Yours sincerely,

Lise Roth

Lise Roth, PhD
Editor
EMBO Molecular Medicine

Photos 400-800 DPI

*Additional important information regarding figures and illustrations can be found at <https://bit.ly/EMBOPressFigurePreparationGuideline>

***** Reviewer's comments *****

Referee #1 (Remarks for Author):

Review of manuscript EMM-2020-13792 - Targeting prominin2 transcription to overcome ferroptosis resistance in cancer by Brown et al.

Ferroptosis is a novel cell death modality that has attracted considerable interest as a potential therapeutic option to treat a series of tumour states and defined lineages that challenge current treatment options. Therefore there is a pressing need to translate this into therapeutic options that could efficiently induce this form of cell death for therapeutic benefit.

In the present manuscript the authors suggest a potential role for the HNE-p38-HSF1-PROM2 axis in suppressing ferroptosis and that this pathway is amenable to pharmacological intervention and could be leveraged to improve ferroptosis inducing strategies.

Having said this in this reviewer opinion the articles has a series of short comings that significantly impact on the authors conclusions and would require attention:

1- The role of HNE in activating p38 is not convincingly demonstrated. For instance, concentrations used here (>20µM) are not at all in the physiological range. More importantly, the level of alkylation seen by the authors (for example as in Fig 1E) argues against any kind of specificity.

2- Additionally, the impact of p38 in ferroptosis is weak and relies mostly on pharmacological data using a single inhibitors (BIRB). Therefore it will be important that the authors provide additional evidence for a role of p38 - for instance by introducing constitutively active variants or generating p38 deficient cell lines.

3- Similarly, a link to HSF1 falls short for the same reason, potentially relevant here is the similar functional groups present in the small molecule used to inhibit HFS1 (KRIBB11) which share a similar NO2 group that can be activated to a potent alkylating agent - similar to the GPX4 inhibitor ML210. The knockdown data is also problematic as this seems to rely on a single siRNA and no rescue control was performed. In figure 3G the effect of RSL3 doesn't seem to be related to a specific action of GPX4 as it is know that selectivity of this drugs is lost at concentration higher than 500nM. Additionaly, given the general toxicity expected by HSF1 loss its not entirely clear if the effect will be due to a specific loss of buffering ferroptosis or simply because the cells are on the verge of dying.

4- Another important consideration would to show that the PROM2 expression, potentially using a doxocycline system to titrate expression to levels similarly to the ones observed upon RSL3

challenge, can rescue the sensitizing effects generated by inhibiting p38 and HSF1.

5- Regarding the in vivo experiments it is not clear if indeed ferroptosis or lipid peroxidation is the major culprit of the effects observed. Critical experiments showing that system Xc- deficient xenografts are more sensitive to manipulation of the p38-HSF1 axis would have been informative as well as assess the impact of ferroptosis inhibitors on the IKE+KRIB11 combination.

Referee #2 (Remarks for Author):

This manuscript by Brown. et al examined the mechanism of induction of promin2 (PROM2) expression, which functions to resist ferroptosis under pro-ferroptotic stress. The results reveal that 4HNE-p38 phosphorylation-HSF1 signaling induces PROM2 transcription, and that pharmacological inhibition of p38 and HSF1 can sensitize cancer cells to ferroptosis. These findings provide a contribution to the mechanism of ferroptosis resistance, with potential implication for the therapy of ferroptosis-resistant cancer. However, I have several comments, in particular i) whether 4HNE is a specific lipid metabolite to induce PROM2, and ii) the mechanism of p38 phosphorylation by 4HNE.

1) Lipid peroxidation is known to generate a variety of lipid oxidation byproducts besides 4-HNE, such as 4HHE, HEL, CRA, 4-ONE, and acrolein etc. Yet, the authors only examined the effects of 4-HNE and MDA, and concluded that 4HNE was a specific lipid metabolite for the PROM2 induction. With that being said, the authors should show whether lipid metabolites other than 4-HNE and MDA may influence p38 phosphorylation and PROM2 induction.

2) The mechanism of increased p38 phosphorylation (p-p38) by 4-HNE remains unclear: Is 4-HNE-adducted p38 more prone to become phosphorylated? Could it be that free 4HNE or 4HNE-adducted upstream molecule(s) promote p38 phosphorylation? Does unphosphorylated p38 fail to bind to 4-HNE? Please clarify these questions.

3) To confirm the proposed signaling for ferroptosis resistance, the effects of p38 and HSF1 inhibitors need to be studied in cells overexpressing PROM2.

4) Additional control groups need to be included for some experiments: Please show the BIRB treatment group without RSL3 in Figure 2C-E and Figure 3C-D. Please show the KRIBB11 treatment group without ferroptosis inducers in Figures 4A-B and 4E. In Fig 4F and 4G, please also examine RSL3 +/- KRIBB11.

5) In the in vivo experiments, why did the authors just examine the effect of HSF1 inhibitor (KRIBB11) and not that of p38 inhibitor (BIRB)?

6) In Fig 6, including the tumor image would be important. In Fig 6D, please add higher power magnification images of IHC to understand the positive localization in the tissues/cells.

7) For the cell survival analysis, other cell death assays are required in addition to the crystal violet staining, because cell death rate was not severe in most results (50-70%). Please include additional data, such as by LDH release and presentation of cell image, in the key experiments.

8) In the KM plotter analysis in supplemental Fig 5, I realized that the authors chose different Affymetrix ID in each set of cancer data. In the data set, the gene symbol PROM2 includes four Affymetrix IDs. The authors used ID of 1562378_s_at in lung cancer and 234198_at in gastric cancer. When I applied a single Affymetrix ID, the tendency of the mortality was not consistent.

Regarding the breast cancer graph, I tried to reproduce it using the KM plotter data; however, all the Affymetrix IDs of PROM2 failed to reproduce the graph shown in the manuscript. Explain this issue and add detailed description in the method or legend for the reproducibility of the results.

9) Typos: In figure 1 legend, 52 uM 4HNE ->25 uM 4HNE?

Referee #3 (Comments on Novelty/Model System for Author):

The technical data is convincing.

No data about animal ethics / in vivo protocol approval is given

Referee #3 (Remarks for Author):

In their manuscript entitled "Targeting prominin2 transcription to overcome ferroptosis resistance in cancer", Caitlin W. Brown et al. worked on cancer cells and they report that 4HNE stimulates PROM2 transcription via an HSF1-dependant mechanism in the context of ferroptosis resistance. The authors showed that 4HNE directly targeted p38MAPK to increase PROM2 mRNA&protein expression levels. In this study, resistance to GPX4 inhibition was obtained by blocking HSF1, both in vitro and in vivo.

The article is nicely written, figures are clear, and results are of interest in the aim of stimulating ferroptosis to tackle cancer proliferation and resistance to anticancer treatments.

The authors used several cell lines to study and describe the cellular mechanism, which is an asset. Although most of the cellular players involved in the signaling pathway of this work (HSF1, 4HNE, GPX4, p38, see Fig7) were already described separately as being involved in ferroptosis, the authors managed to link them into a single pathway/model of ferroptosis resistance, providing a strong rationale for using different combination of inhibitors leading to effective cancer therapy, especially in the context of trying to overcome resistance to ferroptosis.

Main remarks :

- Fig 5D and 5E: Page 9, the authors state "We then evaluated the ability of the parental, sensitive cells (MDA-MB-231S) and resistant (MDA-MB-231R) cells to survive treatment with RSL3 or IKE singly or in combination with KRIBB11 (Figures 5D and E)." The reviewer does not see any data about RSL3 singly or in combination with KRIBB11 in fig 5D and fig 5E, nor in the text.
- in vivo experiments: Please indicate if the study was approved by an independent ethics committee, together with the approval number
- in vivo experiments: please provide the number of mice per group, and how many mice were removed from the study earlier (if applicable)
- Figs 6B and 6C: the axis legend "days post-treatment" is confusing, since it may correspond to days of treatment? Were those tumors measured daily after treatment disruption?
- The reviewer does not understand the treatment sequence (in vivo experiment): in the schematic 6A, treatments start at day 22, until day 35. This is a 13-day period, and this contrasts with the 8-day timescale of Figs 6B and 6C. Please clarify in the text and/or material and methods section.
- Please, provide the origin/clones used for immunohistochemistry used on tumors (if different from those used for the western blots)
- Page 11, the authors state "Notably, 4HNE staining increased with IKE treatment but decreased with the combined treatment compared to the control" Fig 6D does not support this assertion. Did you mean: "Notably, Prominin2 staining increased with IKE treatment but decreased with the combined treatment compared to the control"

Minor remarks:

- Page 7: line 16, please replace RSLE by RSL3
- Suppl. Fig 4C: To highlight the absence of specific effect of dasatinib and erlotinib on ferroptosis the authors plotted the RSL3/dasatinib/KRIBB11 and RSL3/dasatinib/KRIBB11 effects. It seems that there is an effect at 5 and 6 μ M (respectively, which is close to the decrease in survival obtained at 2 μ M for BIRB and KRIBB11). The controls with RSL3/dasatinib and RSL3/erlotinib are lacking to conclude on this decrease.
- Please, provide information (in the Mat&Meth section) on the experiment aiming at obtaining RSL3 resistant MDAMB cells. What was the IC50 of RSL3 before (MDAMB231S) and after (MDAMB231R)?
- Page 10, sentence starting with "Mice were randomized": Please add the condition KRIBB11 alone (50mg/kg) in the sentence, as well as in the schematic Fig 6A
- Fig 7: Nice figure, however no sentence in the text refers to Fig.7. Could you please add a short sentence to recapitulate the underlying mechanism and the original pathway described in this work (in the conclusion section, with reference to fig7)?

EMM-2020-13792

Response to Reviewers' Comments

Reviewer 1

Comment #1: The role of HNE in activating p38 is not convincingly demonstrated. For instance, concentrations used here ($>20\mu\text{M}$) are not at all in the physiological range. More importantly, the level of alkylation seen by the authors (for example as in Fig 1E) argues against any kind of specificity.

Response: We appreciate the reviewer's concern. However, we note the following: 1) There is existing evidence in the literature that 4-HNE activates p38 (see reference 28); 2) We examined the literature and found that physiological range of 4HNE in serum plasma is reported to be as high as $20\mu\text{M}$ (<https://www.ncbi.nlm.nih.gov/pmc/articles/PMC4038367/>). Ferroptotic cells could be assumed to have much higher levels and as such we consider $25\mu\text{M}$ to be physiologically relevant; 3) We consider the data in Figure 1E to be significant given the difference between the EtOH group and the 4HNE/RSL3 groups, as well as the nature of the experiment (immunoblotting for 4HNE).

Comment #2: "Additionally, the impact of p38 in ferroptosis is weak and relies mostly on pharmacological data using a single inhibitor (BIRB). Therefore, it will be important that the authors provide additional evidence for a role of p38 - for instance by introducing constitutively active variants or generating p38 deficient cell lines."

Response: Although BIRB is used frequently as a specific p38 inhibitor, we have now used another specific p38 inhibitor (SB202190) to address the reviewer's concern and observed the same inhibitory effect on *PROM2* mRNA expression. These new data are included in **Figure EV2D**. We believe that extensive characterization of the role of p38 variants in this pathway is unnecessary given the focus of the manuscript.

Comment #3: A link to HSF1 falls short for the same reason, potentially relevant here is the similar functional groups present in the small molecule used to inhibit HFS1 (KRIBB11) which share a similar NO_2 group that can be activated to a potent alkylating agent - similar to the GPX4 inhibitor ML210. The knockdown data is also problematic as this seems to rely on a single siRNA and no rescue control was performed. In figure 3G the effect of RSL3 doesn't seem to be related to a specific action of GPX4 as it is known that selectivity of this drugs is lost at concentration higher than 500nM . Additionally, given the general toxicity expected by HSF1 loss its not entirely clear if the effect will be due to a specific loss of buffering ferroptosis or simply because the cells are on the verge of dying."

Response: In response to the reviewer's comment, we note that KRIBB11 is considered to be a highly specific HSF-1 inhibitor that has been used in many studies. In addition, we substantiated the KRIBB11 data by using a pooled siRNA (and not a single siRNA) to knock-down HSF-1 expression with similar consequences. Given that we used a pooled siRNA, it was not possible to do a rescue experiment. To address the valid concern raised about general toxicity, we have

performed a rescue experiment with ferrostatin-1 to verify that the cells are dying as a result of ferroptosis and not general toxicity. These new data are included in **Figure EV3D**.

Comment #4: Another important consideration would be to show that the PROM2 expression, potentially using a doxycycline system to titrate expression to levels similarly to the ones observed upon RSL3 challenge, can rescue the sensitizing effects generated by inhibiting p38 and HSF1.

Response: To address this concern, we generated MCF10A cells with elevated levels of PROM2 expression and demonstrated that this increase in PROM2 expression has a significant effect on mitigating the effect of inhibiting p38 and HSF1. These new data are included in **Figure EV3C**.

Comment #5: “Regarding the *in vivo* experiments it is not clear if indeed ferroptosis or lipid peroxidation is the major culprit of the effects observed. Critical experiments showing that system Xc- deficient xenografts are more sensitive to manipulation of the p38-HSF1 axis would have been informative as well as assess the impact of ferroptosis inhibitors on the IKE+KRIB11 combination.”

Response: We do not believe that an *in vivo* experiment with xCT-deficient xenografts would add to the manuscript because the Stockwell group has shown that IKE is a selective and stable inhibitor of system xCT and that it *induces ferroptosis* (see reference 5). They also reported its efficacy *in vivo* for inducing ferroptosis and slowing tumor growth (see reference 5). Our data add to these findings by demonstrating that HSF1 inhibition *in vivo* sensitizes resistant cells to IKE.

Reviewer 2

Comment #1: “Lipid peroxidation is known to generate a variety of lipid oxidation byproducts besides 4-HNE, such as 4HHE, HEL, CRA, 4-ONE, and acrolein etc. Yet, the authors only examined the effects of 4-HNE and MDA, and concluded that 4HNE was a specific lipid metabolite for the PROM2 induction. With that being said, the authors should show whether lipid metabolites other than 4-HNE and MDA may influence p38 phosphorylation and PROM2 induction.”

Response: In response to this comment, we emphasize that we compared two reactive lipid species that are associated with ferroptosis (4HNE and MDA) and concluded that 4HNE induces PROM2 and that MDA does not. We are not concluding that other lipid reactive species may not induce PROM2 but that PROM2 can be induced by a specific lipid reactive species (4HNE). We consider this finding to be novel and significant.

Comment #2: “The mechanism of increased p38 phosphorylation (p-p38) by 4-HNE remains unclear: Is 4-HNE-adducted p38 more prone to become phosphorylated? Could it be that free 4HNE or 4HNE-adducted upstream molecule(s) promote p38 phosphorylation? Does unphosphorylated p38 fail to bind to 4-HNE? Please clarify these questions.”

Response: We apologize for forgetting to cite a study demonstrating that 4HNE targets p38 and induces activation via phosphorylation, which now cite in the revised manuscript²⁸. Essentially, the authors reported that p38 is activated by 4HNE because it complexes with p38, which results

in p38 phosphorylation. The mechanism by which 4HNE activates its target kinases uses Michael additions.

Comment #3: “To confirm the proposed signaling for ferroptosis resistance, the effects of p38 and HSF1 inhibitors need to be studied in cells overexpressing PROM2.

Response: To address this concern, we generated MCF10A cells with elevated levels of PROM2 expression and demonstrated that this increase in PROM2 expression has a significant effect on mitigating the effect of inhibiting p38 and HSF1. These new data are included in **Figure EV3C**.

Comment #4: “Additional control groups need to be included for some experiments. Please show the BIRB treatment group without RSL3 in Figure 2C-E and Figure 3C-D. Please show the KRIBB11 treatment group without ferroptosis inducers in Figures 4A-B and 4E. In Fig 4F and 4G, please also examine RSL3 +/- KRIBB11.”

Response: We have addressed all of the issues in Comment #4 as follows: 1) We observed that BIRB alone did not induce PROM2 expression. These new data are included in **Figure EV2C**; 2) We examined the effect of KRIBB11 alone and observed that neither *PROM2* nor *HSF1* mRNA levels were significantly altered in response to KRIBB11 treatment. These new data are included in **Figure EV4D** and **E**; 3) We observed that RSL3 + KRIB11 did not decrease the viability of either S1 or HMLE cells. These new data are included in **Figure EV4F** and **EV4G**.

Comment #5: “In the in vivo experiments, why did the authors just examine the effect of HSF1 inhibitor (KRIBB11) and not that of p38 inhibitor (BIRB)?”

Response: The reviewer raises an interesting point, but we wanted to directly test the inhibition of HSF1 in cells experiencing ferroptotic stress. By inhibiting HSF1 directly we reasoned that there would be fewer off-target effects than by inhibiting an upstream kinase involved in many signaling pathways. We were also concerned about the use of a systemic *in vivo* pan-p38 inhibitor in combination with IKE and how that would affect the tolerability of the drug combination.

Comment #6: “In Fig 6, including the tumor image would be important. In Fig 6D, please add higher power magnification images of IHC to understand the positive localization in the tissues/cells.”

Response: In response, we have added higher magnification insets images to **Figure 6D**.

Comment #7: “In the KM plotter analysis in supplementary Fig 5, I realized that the authors chose different Affymetrix ID in each set of cancer data. In the data set, the gene symbol PROM2 includes four Affymetrix IDs. The authors used ID of 1562378_s_at in lung cancer and 234198_at in gastric cancer. When I applied a single Affymetrix ID, the tendency of the mortality was not consistent. Regarding the breast cancer graph, I tried to reproduce it using the KM plotter data; however, all the Affymetrix IDs of PROM2 failed to reproduce the graph shown in the manuscript. Explain this issue and add detailed description in the method or legend for the reproducibility of the results.”

Response: We thank the reviewer for noting our mistake. In response, we have removed these data from the paper.

Comment #8: Minor comment: Typos: In figure 1 legend, 52 uM 4HNE ->25 uM 4HNE

Response: The typo has been corrected.

Reviewer 3

Comment #1: “Fig 5D and 5E: Page 9, the authors state "We then evaluated the ability of the parental, sensitive cells (MDA-MB-231S) and resistant (MDA-MB-231R) cells to survive treatment with RSL3 or IKE singly or in combination with KRIBB11 (Figures 5D and E)." The reviewer does not see any data about RSL3 singly or in combination with KRIBB11 in fig 5D and fig 5E, nor in the text. “

Response: In response, we have now performed the experiment with RSL3, and the data are included in **Figure EV5A** and **EV5B**.

Comment #2: “The reviewer does not understand the treatment sequence (in vivo experiment): in the schematic 6A, treatments start at day 22, until day 35. This is a 13-day period, and this contrasts with the 8-day timescale of Figs 6B and 6C. Please clarify in the text and/or material and methods section.”

Response: We thank the reviewer for noting this mistake, which has been corrected in the Figure 6A schematic.

Minor Comments:

“in vivo experiments: Please indicate if the study was approved by an independent ethics committee, together with the approval number”

“in vivo experiments: please provide the number of mice per group, and how many mice were removed from the study earlier (if applicable)”

Response: This information has now been included the Methods section.

“Figs 6B and 6C: the axis legend "days post-treatment" is confusing, since it may correspond to days of treatment? Were those tumors measured daily after treatment disruption? “

Response: We have clarified the labeling of the axis.

22nd Apr 2021

Dear Dr. Mercurio,

Thank you for the submission of your revised manuscript to EMBO Molecular Medicine. We have now received feedback from the two referees who re-reviewed your manuscript. As you will see from the reports below, both referees acknowledge your efforts to address their initial concerns, and recognize that the manuscript has significantly improved. However, they also both raise issues that remain unanswered.

As EMBO Press encourages a single round of major revisions only, we would normally reject the manuscript at this stage. However, as both reviewers recognize (as we do) the interest of the study, we would like to exceptionally invite a second round of revisions. Please be aware that this will be the last chance for you to address the points raised by the referees. As indicated below, we do not ask for experimental validation of all the referees' points. More specifically:

Referee #1:

Comment #1: please address this comment in writing, both in the rebuttal and in the manuscript.
Comment #2: if you have data at hand (genetic evidence), we will be happy for you to include it, however experimental data will not be required for further consideration of the manuscript.
Comment #3: please address this point in writing, both in the rebuttal and in the manuscript.
Comment #4: please include the whole range of concentrations and discuss the use of MCF10A.
Comment #5: we do not ask you to provide additional xenograft experiments. Please discuss.

Referee #2:

Comment #1: please address this point experimentally.
Comment #2: please discuss.
Comment #3: please clarify.
Please also address the minor points from this referee.

Please also address the following editorial issues:

- Please upload your figures separately, one file/one page per figure (including EV figures).
- Please provide up to 5 keywords.
- Please place the Material and Methods section after the Discussion
- Primary datasets produced in this study need to be deposited in an appropriate public database and their accession number should be listed in a Data Availability Section. If no new data was generated, please indicate: "This study includes no data deposited in external repositories"
- The Paper Explained: Please provide a summary of your article highlighting the medical issue you are addressing, the results obtained and their clinical impact. This may be edited to ensure that readers understand the significance and context of the research. Please refer to any of our published articles for an example.
- Please make sure the funding information provided in the manuscript match those provided in the submission system.
- Please reformat the references to have 10 authors listed before et al, and in alphabetical order.

- Please make sure all figures are referenced in the main text.
- As part of the EMBO Publications transparent editorial process initiative (see our Editorial at <http://embomolmed.embopress.org/content/2/9/329>), EMBO Molecular Medicine will publish online a Review Process File (RPF) to accompany accepted manuscripts.

In the event of acceptance, this file will be published in conjunction with your paper and will include the anonymous referee reports, your point-by-point response and all pertinent correspondence relating to the manuscript. Let us know whether you agree with the publication of the RPF and as here, if you want to remove or not any figures from it prior to publication.

I look forward to receiving your revised manuscript.

With kind regards,

Lise Roth

Lise Roth, PhD
Editor
EMBO Molecular Medicine

Photos 400-800 DPI

*Additional important information regarding figures and illustrations can be found at <https://bit.ly/EMBOPressFigurePreparationGuideline>

***** Reviewer's comments *****

Referee #1 (Remarks for Author):

Comments to the replies presented for the initial review round of manuscript EMM-2020-13792
First of all I would like to thank the authors for their efforts to reply and try to accommodate the initial concerns/comments raised. We are all aware of the difficulties to approach this during this challenging times.

Comment #1: The role of HNE in activating p38 is not convincingly demonstrated. For instance, concentrations used here ($>20\mu\text{M}$) are not at all in the physiological range. More importantly, the level of alkylation seen by the authors (for example as in Fig 1E) argues against any kind of specificity.

Response: We appreciate the reviewer's concern. However, we note the following: 1) There is existing evidence in the literature that 4-HNE activates p38 (see reference 28); 2) We examined the literature and found that physiological range of 4HNE in serum plasma is reported to be as high as $20\mu\text{M}$ (<https://www.ncbi.nlm.nih.gov/pmc/articles/PMC4038367/>). Ferroptotic cells could be assumed to have much higher levels and as such we consider $25\mu\text{M}$ to be physiologically relevant; 3) We consider the data in Figure 1E to be significant given the difference between the EtOH group and the 4HNE/RSL3 groups, as well as the nature of the experiment (immunoblotting for 4HNE).

Reply: The evidence mentioned by the author, mostly the citations within the cited reference, are by no means unequivocal. Measuring 4-HNE levels unambiguously is rather complicated, given the reactive nature of this electrophile. Overall, the $20\mu\text{M}$ range of free 4-HNE appears to be exaggerated. Having said this, the scope of this discussion potentially falls out of the current work but the manuscript would benefit from highlighting these potential shortcomings. Finally, Figure 1E is far from showing or supporting specificity despite what the authors highlight. There might be indeed an issue with the detection method and more refined techniques, including proteomic mapping of 4-HNE adducted protein. The comparison with RSL3 would have helped clarify something similar that has been tried in the past (Quantitative Profiling of Protein Carbonylation in Ferroptosis by an Aniline-Derived Probe. Chen Y, Liu Y, Lan T, Qin W, Zhu Y, Qin K, Gao J, Wang H, Hou X, Chen N, Friedmann Angeli JP, Conrad M, Wang C. J Am Chem Soc. 2018 Apr 4;140(13):4712-4720.). Yet in that study the notion of lack of specificity is also evident, and furthermore, p38 was not found as a potential target of endogenously generated electrophiles and also given the high. Again, these differences between experimental conditions are complex, but I would also encourage them to tune their discussion based on this recognition.

Comment #2: "Additionally, the impact of p38 in ferroptosis is weak and relies mostly on pharmacological data using a single inhibitor (BIRB). Therefore, it will be important that the authors provide additional evidence for a role of p38 - for instance by introducing constitutively active variants or generating p38 deficient cell lines."

Response: Although BIRB is used frequently as a specific p38 inhibitor, we have now used another specific p38 inhibitor (SB202190) to address the reviewer's concern and observed the same inhibitory effect on PROM2 mRNA expression. These new data are included in Figure EV2D. We believe that extensive characterization of the role of p38 variants in this pathway is unnecessary given the focus of the manuscript.

Reply: The authors provide data with an additional inhibitor, which strengthens their point, yet the lack of genetic evidence makes the link weak. In this reviewer assessment, the role of p38 is not out of the manuscript focus, and the experiments suggested would not have required substantial time

or resources.

Comment #3: A link to HSF1 falls short for the same reason, potentially relevant here is the similar functional groups present in the small molecule used to inhibit HSF1 (KRIBB11) which share a similar NO₂ group that can be activated to a potent alkylating agent - similar to the GPX4 inhibitor ML210. The knockdown data is also problematic as this seems to rely on a single siRNA and no rescue control was performed. In figure 3G the effect of RSL3 doesn't seem to be related to a specific action of GPX4 as it is known that selectivity of this drug is lost at concentration higher than 500nM. Additionally, given the general toxicity expected by HSF1 loss it's not entirely clear if the effect will be due to a specific loss of buffering ferroptosis or simply because the cells are on the verge of dying."

Response: In response to the reviewer's comment, we note that KRIBB11 is considered to be a highly specific HSF-1 inhibitor that has been used in many studies. In addition, we substantiated the KRIBB11 data by using a pooled siRNA (and not a single siRNA) to knock-down HSF-1 expression with similar consequences. Given that we used a pooled siRNA, it was not possible to do a rescue experiment. To address the valid concern raised about general toxicity, we have performed a rescue experiment with ferrostatin-1 to verify that the cells are dying as a result of ferroptosis and not general toxicity. These new data are included in Figure EV3D.

Reply: Thank you to the authors for clarifying this - to this reviewer understanding, the lack of a rescue experiment undermines these conclusions' strength and should be discussed accordingly. Additionally, the authors might want to acknowledge that the rescue of Fer-1 is only partial and that there might be non-ferroptosis related toxicity/cell death taking place.

Comment #4: Another important consideration would be to show that the PROM2 expression, potentially using a doxycycline system to titrate expression to levels similarly to the ones observed upon RSL3 challenge, can rescue the sensitizing effects generated by inhibiting p38 and HSF1.

Response: To address this concern, we generated MCF10A cells with elevated levels of PROM2 expression and demonstrated that this increase in PROM2 expression has a significant effect on mitigating the effect of inhibiting p38 and HSF1. These new data are included in Figure EV3C.

Reply: Could the author present the whole range of concentrations as done for the other experiments. This single point experiment indicates that forced expression of PROM2 only partially protects cells from the combination of drugs. Also, the overexpression of PROM2 in the MCF10A cells seems like a bad choice since this cell line's is highly resistant to ferroptosis even when both compounds are combined.

Comment #5: "Regarding the in vivo experiments it is not clear if indeed ferroptosis or lipid peroxidation is the major culprit of the effects observed. Critical experiments showing that system Xc- deficient xenografts are more sensitive to manipulation of the p38-HSF1 axis would have been informative as well as assess the impact of ferroptosis inhibitors on the IKE+KRIB11 combination."

Response: We do not believe that an in vivo experiment with xCT-deficient xenografts would add to the manuscript because the Stockwell group has shown that IKE is a selective and stable inhibitor of system xCT and that it induces ferroptosis (see reference 5). They also reported its efficacy in vivo for inducing ferroptosis and slowing tumor growth (see reference 5). Our data add to these findings by demonstrating that HSF1 inhibition in vivo sensitizes resistant cells to IKE.

Reply: Here, I would like to point to the authors that in reference 5, IKE was only shown to drive ferroptosis in vitro unequivocally. In vivo and unequivocal evidence is lacking. For instance, there is no experiment showing that lipid peroxidation contributes to decreased tumour growth, which would characterize ferroptosis. Moreover, since it is known that system Xc- is not required for cysteine uptake (in the form of cystine) in vivo, given that Xct knockout mice are fully viable, the question still stands. Does IKE drive cell death by inhibiting system Xc- and drive ferroptosis in

vivo? Therefore, despite cumbersome, the xenograft would have helped to answer this question and ultimately provide evidence for this. Ultimately, the same applies here, does the combination of IKE+KRIB11 induces ferroptosis in vivo? This is not to discredit the practical impact of the combination but to highlight that this is not unequivocally shown, and therefore the authors should acknowledge these limitations in their discussion.

Referee #2 (Remarks for Author):

The authors appropriately responded most of the reviewer's comments. However, the following questions remains.

1. Although the finding of PROM2 induction by 4HNE is novel as the authors responded, the authors only examined the effects of two lipid oxidation byproducts. Thus, to examine whether other lipid reactive species (such as 4HHE, HEL, CRA, and acrolein) can induce the p38-HSF1-PROM2 signaling is still recommended. The compounds are commercially available and the additional analysis would not be difficult to do.

2. In Figure EV3C, KRIBB1 still enhanced the sensitivity of RSL3 despite the PROM2 overexpression independent of p38-HSF1 signaling. Please explain the reason of this finding. Also, please add the WB data showing overexpression of PROM2 in MCF10A.

"Increased expression of prominin2 in MCF10A cells diminished their sensitivity to p38 and HSF1 inhibition (Figure EV3C)."

This sentence is not correct because the authors did not statistically compare the survival rate between mock cells and overexpression cells.

3. In figure 3C, the figure said "siHSF", but the legend said "siPromin2"? Please confirm.

Minor

In Figure EV3B, an unnecessary word of "60" was overlapped.

In the legend of figure EV3, "5 mM of RSL3" and "2 mM ferrostatin-1" would be incorrect unit.

I would appreciate your advice in responding to two of the comments of Reviewer #1, which you asked us to address in writing.

Comment #1: "The evidence mentioned by the author, mostly the citations within the cited reference, are by no means unequivocal. Measuring 4-HNE levels unambiguously is rather complicated, given the reactive nature of this electrophile. Overall, the 20 μ M range of free 4-HNE appears to be exaggerated. Having said this, the scope of this discussion potentially falls out of the current work but the manuscript would benefit from highlighting these potential shortcomings. Finally, Figure 1E is far from showing or supporting specificity despite what the authors highlight. There might be indeed an issue with the detection method and more refined techniques, including proteomic mapping of 4-HNE adducted protein. The comparison with RSL3 would have helped clarify something similar that has been tried in the past (Quantitative Profiling of Protein Carbonylation in Ferroptosis by an Aniline-Derived Probe. Chen Y, Liu Y, Lan T, Qin W, Zhu Y, Qin K, Gao J, Wang H, Hou X, Chen N, Friedmann Angeli JP, Conrad M, Wang C. J Am Chem Soc. 2018 Apr 4;140(13):4712-4720.). Yet in that study the notion of lack of specificity is also evident, and furthermore, p38 was not found as a potential target of endogenously generated electrophiles and also given the high. Again, these differences between experimental conditions are complex, but I would also encourage them to tune their discussion based on this recognition".

The reviewer suggests that the 20 μ M concentration of 4-HNE may be exaggerated but admits that this issue potentially falls out of the scope of the paper and indicates that is difficult to assess the appropriate concentration. We are unsure how to modify the text.

From our perspective, the data in Fig. 1E demonstrate specificity, especially with the EtOH control.

Also, the reviewer states that the differences between experimental conditions are complex but asks us to tune our discussion.

Again, we are unsure how to modify the text.

Comment # 5: Here, I would like to point to the authors that in reference 5, IKE was only shown to drive ferroptosis in vitro unequivocally. In vivo and unequivocal evidence is lacking. For instance, there is no experiment showing that lipid peroxidation contributes to decreased tumour growth, which would characterize ferroptosis. Moreover, since it is known that system Xc⁻ is not required for cysteine uptake (in the form of cystine) in vivo, given that Xct knockout mice are fully viable, the question still stands. Does IKE drives cell death by inhibiting system Xc⁻ and drives ferroptosis in vivo? Therefore, despite cumbersome, the xenograft would have helped to answer this question and ultimately provide evidence for this. Ultimately, the same applies here, does the combination of IKE+KRIB11 induces ferroptosis in vivo? This is not to discredit the practical impact of the combination but to highlight that this is not unequivocally shown, and therefore the authors should acknowledge these limitations in their discussion.

The Stockwell paper (ref 5) that the reviewer mentions concludes that IKE induces

ferroptosis in vivo. Although the data may not be as rigorous as the reviewer would like, the data in this paper demonstrate that IKE can inhibit tumor growth in vivo by a mechanism that involves ferroptosis. We add to these findings by demonstrating that tumor that are resistant to IKE can be sensitized by co-treatment with the HSF-1 inhibitor.

We are not sure how to respond with diminishing the impact of the Stockwell paper and our own work.

I have now received the feedback from referee #1, who stated: "regarding comment 1; they lack evidence for an unequivocal and specific role for 4-HNE, so I would suggest that the discussion is modified and the prominent role for 4-HNE be less explicit. One way to approach this would be to justify the effects are derived from lipid peroxidation byproducts such as 4-HNE (but also others!)"

regarding comment 5; the problem is that triggering ferroptosis in vitro and in vivo is very different, particularly when using system Xc- inhibitors. This is exemplified for instance by the phenotype of system Xc- KO mice, which are fully viable but cells derived from these mice readily die in culture. And the sole reason is, cystine is not a major source for cysteine in vivo, it is in vitro though.

So when the authors say that their drug combination triggers ferroptosis in vivo using a system Xc- inhibitor, which is expected to drive cysteine deprivation I don't see the evidence for this. Currently, there is not enough evidence showing that IKE has an on-target (Xct) activity in vivo, even it would have its is very likely that in vivo inhibition of system Xc- would not lead to cysteine starvation, simply because this is not how cells acquired cysteine in vivo. I just would like to state again, the same evidence is missing in the paper they cite from the Stockwell group. Yet, its obvious that the IKE is doing something in vivo as reported by both works - but none actually provided evidence that this is via system Xc- or ferroptosis (again, in vivo). Overall for the field, this is something important, otherwise, many more people might be misled to use IKE as an in vivo ferroptosis-inducing agent - leading to pointless waste of mice and money."

EMM-2020-13792R2**Response to Reviewers' Comments**

We thank the reviewers for their rapid review of our revised manuscript and their thoughtful. Our responses to their additional concerns are below and are based on the Editor's guidance.

Reviewer 1

Comment #1: The evidence mentioned by the author, mostly the citations within the cited reference, are by no means unequivocal. Measuring 4-HNE levels unambiguously is rather complicated, given the reactive nature of this electrophile. Overall, the 20 μ M range of free 4-HNE appears to be exaggerated. Having said this, the scope of this discussion potentially falls out of the current work but the manuscript would benefit from highlighting these potential shortcomings. Finally, Figure 1E is far from showing or supporting specificity despite what the authors highlight. There might be indeed an issue with the detection method and more refined techniques, including proteomic mapping of 4-HNE adducted protein. The comparison with RSL3 would have helped clarify something similar that has been tried in the past (Quantitative Profiling of Protein Carbonylation in Ferroptosis by an Aniline-Derived Probe. Chen Y, Liu Y, Lan T, Qin W, Zhu Y, Qin K, Gao J, Wang H, Hou X, Chen N, Friedmann Angeli JP, Conrad M, Wang C. J Am Chem Soc. 2018 Apr 4;140(13):4712-4720.). Yet in that study the notion of lack of specificity is also evident, and furthermore, p38 was not found as a potential target of endogenously generated electrophiles and also given the high. Again, these differences between experimental conditions are complex, but I would also encourage them to tune their discussion based on this recognition.

Response: In a new experiment, we observed two other lipid peroxidation products (4ONE and 4HHE) were unable to induce prominin 2 expression, arguing for the specificity of 4HNE. Nonetheless, in response to our helpful email dialogue with the reviewer (through the Editor), we have modified the Discussion to state that we can't exclude the possibility that other lipid peroxidation products contribute to the induction of prominin 2 expression.

Comment #2: The authors provide data with an additional inhibitor, which strengthens their point, yet the lack of genetic evidence makes the link weak. In this reviewer assessment, the role of p38 is not out of the manuscript focus, and the experiments suggested would not have required substantial time or resources.

Response: Editor stated that genetic evidence is not needed.

Comment #3: Thank you to the authors for clarifying this - to this reviewer understanding, the lack of a rescue experiment undermines these conclusions' strength and should be discussed accordingly. Additionally, the authors might want to acknowledge that the rescue of Fer-1 is only partial and that there might be non-ferroptosis related toxicity/cell death taking place.

Response: We have modified the results on p. 8 describing Fig. EV3D as requested by the reviewer.

Comment #4: Could the author present the whole range of concentrations as done for the other experiments. This single point experiment indicates that forced expression of PROM2 only

partially protects cells from the combination of drugs. Also, the overexpression of PROM2 in the MCF10A cells seems like a bad choice since this cell line's is highly resistant to ferroptosis even when both compounds are combined.

Response: These data are now provided in Extended Fig. 3D.

Comment #5: Here, I would like to point to the authors that in reference 5, IKE was only shown to drive ferroptosis *in vitro* unequivocally. *In vivo* and unequivocal evidence is lacking. For instance, there is no experiment showing that lipid peroxidation contributes to decreased tumour growth, which would characterize ferroptosis. Moreover, since it is known that system Xc- is not required for cysteine uptake (in the form of cystine) *in vivo*, given that Xct knockout mice are fully viable, the question still stands. Does IKE drives cell death by inhibiting system Xc- and drives ferroptosis *in vivo*? Therefore, despite cumbersome, the xenograft would have helped to answer this question and ultimately provide evidence for this. Ultimately, the same applies here, does the combination of IKE+KRIB11 induces ferroptosis *in vivo*? This is not to discredit the practical impact of the combination but to highlight that this is not unequivocally shown, and therefore the authors should acknowledge these limitations in their discussion.

Response: We thank the reviewer for clarifying this comment in an email dialogue (through the Editor). In response, we have modified and expanded the Discussion to mention that the ability of IKE to induce ferroptosis *in vivo* has not been unequivocally demonstrated and that we can't make this definitive conclusion from our data.

Reviewer 2

Comment #1: Although the finding of PROM2 induction by 4HNE is novel as the authors responded, the authors only examined the effects of two lipid oxidation byproducts. Thus, to examine whether other lipid reactive species (such as 4HHE, HEL, CRA, and acrolein) can induce the p38-HSF1-PROM2 signaling is still recommended. The compounds are commercially available and the additional analysis would not be difficult to do.

Response: In a new experiment, we observed that two other lipid peroxidation products (4ONE and 4HHE) were unable to induce prominin 2 expression, arguing for the specificity of 4HNE. These data are now provided in Extended Fig. 1B.

Comment #2: In Figure EV3C, KRIBB1 still enhanced the sensitivity of RSL3 despite the PROM2 overexpression independent of p38-HSF1 signaling. Please explain the reason of this finding. Also, please add the WB data showing overexpression of PROM2 in MCF10A. "Increased expression of prominin2 in MCF10A cells diminished their sensitivity to p38 and HSF1 inhibition (Figure EV3C)."

This sentence is not correct because the authors did not statistically compare the survival rate between mock cells and overexpression cells.

Response: We have provided the WB requested in Extended Fig. 3C. We have also repeated the experiment with the PROM2 expressing cells using a range of RSL3 concentrations (Extended

Fig. 3D). The difference in the survival rate between the mock and overexpression cells is statistically significant as we now indicate on the figure.

Comment #3: Minor Comments

Response: Corrected

Comment #4: In figure 3C, the figure said "siHSF", but the legend said "siPromin2"? Please confirm.

Response: Corrected

Comment #5: Minor

In Figure EV3B, an unnecessary word of "60" was overlapped.

In the legend of figure EV3, "5 mM of RSL3" and "2 mM ferrostatin-1" would be incorrect unit.

Response: Corrected

21st May 2021

Dear Dr. Mercurio,

Thank you for the submission of your revised manuscript to EMBO Molecular Medicine. We have now received the enclosed reports from the referees who re-reviewed your manuscript. As you will see, they are supportive of publication, and I am therefore pleased to inform you that we will be able to accept your manuscript once the following editorial points will be addressed:

1/ Main manuscript text:

- Please answer/correct the changes suggested by our data editors in the main manuscript file attached (in track changes mode). Please use this file for any further modification.
- Please remove the highlights in the text.
- Material and methods:
 - o Cells: please indicate whether cells were authenticated and tested for mycoplasma contamination. Please indicate the culture conditions.
 - o Antibodies: please provide the dilutions used in your experiments.
 - o siRNA: please provide references.
 - o Mice: please indicate the origin of the mice, and their housing and husbandry conditions.

2/ Figures:

- Please make sure all figures are referenced in the main text (callouts are missing for Fig 5B-E, Fig 7)
- Statistics: please indicate in the legends or in the figures the exact n= and exact p= values, not a range, along with the statistical test used. Some people found that to keep the figures clear, providing a supplemental table with all exact p-values was preferable. You are welcome to do this if you want to.

3/ We would also encourage you to include the source data for figure panels that show essential data. Numerical data should be provided as individual .xls or .csv files (including a tab describing the data). For blots or microscopy, uncropped images should be submitted (using a zip archive if multiple images need to be supplied for one panel). Additional information on source data and instruction on how to label the files are available at

4/ Checklist:

Please fill in section B/2 and B/5.

5/ Thank you for providing a synopsis text and figure. I slightly edited the text to fit our style and format, please let me know if you agree with the following:

Stimulating ferroptosis has emerged as a potential therapeutic strategy against cancer. Some tumor cells, however, are resistant to known ferroptosis stimuli. This study identifies mechanisms that contribute to ferroptosis and develops strategies to overcome resistance.

- Prominin 2 expression was induced by ferroptotic stimuli and contributed to resistance.
- PROM2 transcription was stimulated by the lipid metabolite 4-hydroxynonenal (4-HNE) via a mechanism involving p38 MAP-kinase-mediated activation of heat shock factor 1 (HSF1).
- Resistant tumor cells were sensitized to drugs that induce ferroptosis by HSF1 inhibition.

- Cytostasis of established tumors in mice was observed upon combination treatment with a ferroptosis- inducing drug and an HSF1 inhibitor.

6/ As part of the EMBO Publications transparent editorial process initiative (see our Editorial at <http://embomolmed.embopress.org/content/2/9/329>), EMBO Molecular Medicine will publish online a Review Process File (RPF) to accompany accepted manuscripts.

This file will be published in conjunction with your paper and will include the anonymous referee reports, your point-by-point response and all pertinent correspondence relating to the manuscript. Let us know whether you agree with the publication of the RPF and as here, if you want to remove or not any figures from it prior to publication.

I look forward to receiving your revised manuscript.

Yours sincerely,

Lise Roth

Lise Roth, PhD
Editor
EMBO Molecular Medicine

Photos 400-800 DPI

*Additional important information regarding figures and illustrations can be found at <https://bit.ly/EMBOPressFigurePreparationGuideline>

The system will prompt you to fill in your funding and payment information. This will allow Wiley to send you a quote for the article processing charge (APC) in case of acceptance. This quote takes into account any reduction or fee waivers that you may be eligible for. Authors do not need to pay any fees before their manuscript is accepted and transferred to our publisher.

***** Reviewer's comments *****

Referee #1 (Remarks for Author):

I have no further remarks and would like to praise the authors for their work.

Referee #2 (Remarks for Author):

The manuscript was revised appropriately.

The authors performed the requested changes.

7th Jun 2021

We are pleased to inform you that your manuscript is accepted for publication and is now being sent to our publisher to be included in the next available issue of EMBO Molecular Medicine.

Corresponding Author Name: Arthur Mercurio

Manuscript Number: EMM-2020-13792